# Temporal-Difference Variational Continual Learning

## Abstract

A crucial capability of Machine Learning models in real-world applications is the ability to continuously learn new tasks. This adaptability allows them to respond to potentially inevitable shifts in the data-generating distribution over time. However, in Continual Learning (CL) settings, models often struggle to balance learning new tasks (plasticity) with retaining previous knowledge (memory stability). Consequently, they are susceptible to Catastrophic Forgetting, which degrades performance and undermines the reliability of deployed systems. Variational Continual Learning methods tackle this challenge by employing a learning objective that recursively updates the posterior distribution and enforces it to stay close to the latest posterior estimate. Nonetheless, we argue that these methods may be ineffective due to compounding approximation errors over successive recursions. To mitigate this, we propose new learning objectives that integrate the regularization effects of multiple previous posterior estimations, preventing individual errors from dominating future posterior updates and compounding over time. We reveal insightful connections between these objectives and Temporal-Difference methods, a popular learning mechanism in Reinforcement Learning and Neuroscience. We evaluate the proposed objectives on challenging versions of popular CL benchmarks, demonstrating that they outperform standard Variational CL methods and non-variational baselines, effectively alleviating Catastrophic Forgetting.

## 1 Introduction

A fundamental aspect of robust Machine Learning (ML) models is to learn from non-stationary sequential data. In this scenario, two main properties are necessary: first, models must learn from new incoming data — potentially from a different task -– with satisfactory asymptotic performance and sample complexity. This capability is called plasticity. Second, they must retain the knowledge from previously learned tasks, know as memory stability. When this does not happen, and the performance of previous tasks degrades, the model suffers from Catastrophic Forgetting (Goodfellow et al., 2015; McCloskey & Cohen, 1989). These two properties are the central core of Continual Learning (CL) (Schlimmer & Fisher, 1986; Abraham & Robins, 2005), being strongly relevant for ML systems susceptible to test-time distributional shifts.

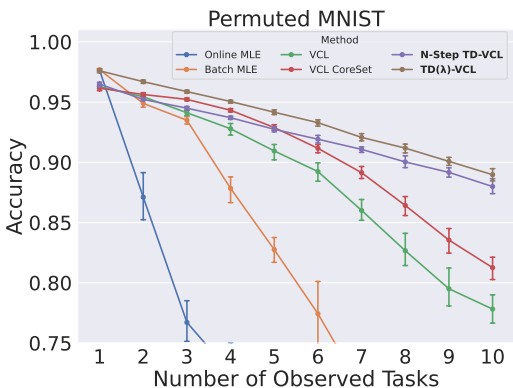

Figure 1: **Average accuracy across observed tasks in the PermutedMNIST benchmark**. The TD-VCL approach, proposed in this work, leads to a substantial improvement against standard VCL and non-variational approaches.

Given the critical importance of this topic, extensive literature addresses the challenges of CL in traditional ML methods (Schlimmer & Fisher, 1986; Sutton & Whitehead, 1993; McCloskey & Cohen, 1989; French, 1999) and, more recently, for overparameterized models (Hadsell et al., 2020; Goodfellow et al., 2015; Serra et al., 2018). Particularly in this work, we focus on the Bayesian setting, as we argue that it provides a princi-

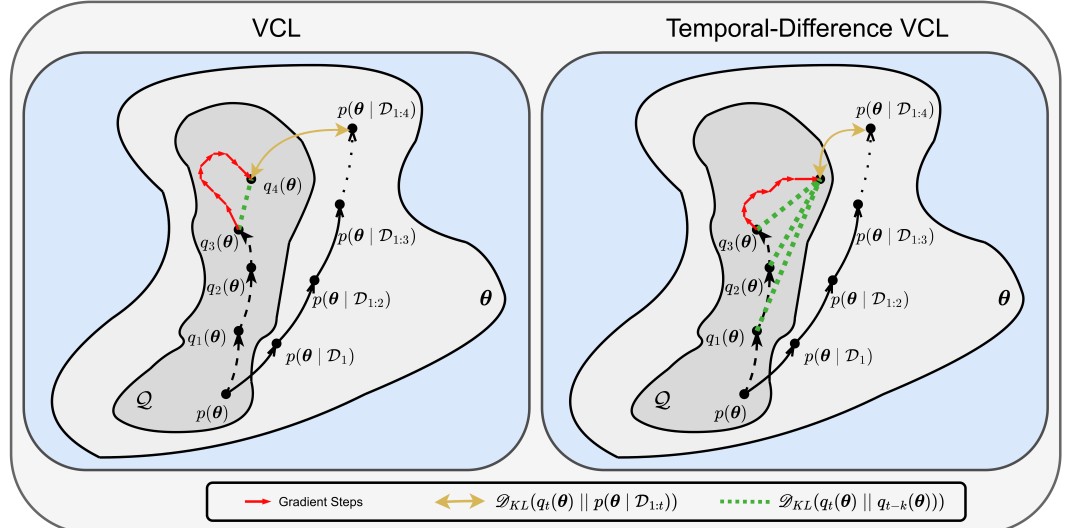

Figure 2: **An intuitive illustration of how TD-VCL functions in comparison to vanilla VCL**. At each timestep $t$, a new task dataset $\mathcal{D}_t$ arrives. Both methods aim to learn variational parameters $q_t(\boldsymbol{\theta})$ over a family of distributions $\mathcal{Q}$ that approximates the true posterior $p(\boldsymbol{\theta} \mid \mathcal{D}_{1:t})$ via minimizing the KL divergence $\mathcal{D}_{KL}(q_t(\boldsymbol{\theta}) \mid\mid p(\boldsymbol{\theta} \mid \mathcal{D}_{1:t}))$. VCL optimization (left) is only constrained by the most recent posterior, which compounds approximation errors from previous estimations and potentially deviates far from the true posterior. TD-VCL (right) is regularized by a sequence of past estimations, alleviating the impact of compounded errors.

pled framework for learning in online or low-data regimes. We investigate the Variational Continual Learning (VCL) approach (Nguyen et al., 2018). As detailed in Section 3, VCL identifies a recursive relationship between subsequent posterior distributions over tasks. A variational optimization objective then leverages this recursion, which regularizes the updated posterior to stay close to the very latest posterior approximation. Nevertheless, we argue that solely relying on a single previous posterior estimate for building up the next optimization target may be ineffective, as the approximation error propagates to the next update and compounds after successive recursions. If a particular estimation is especially poor, the error will be carried over to the next step entirely, which can dramatically degrade model's performance.

In this work, we show that the same optimization objective can be represented as a function of a sequence of previous posterior estimates and task likelihoods. We thus propose a new Continual Learning objective, n-Step KL VCL, that explicitly regularizes the posterior update considering several past posterior approximations. By considering multiple previous estimates, the objective dilutes individual errors, allows correct posterior approximates to exert a corrective influence, and leverages a broader global context to the learning target, reducing the impact of compounding errors over time. Figure 2 illustrates the underlying mechanism.

We further generalize this unbiased optimization target to a broader family of CL objectives, namely Temporal-Difference VCL, which constructs the learning target by prioritizing the most recent approximated posteriors. We reveal a link between the proposed objective and Temporal-Difference (TD) methods, a popular learning mechanism in Reinforcement Learning (Sutton, 1988) and Neuroscience (Schultz et al., 1997). Furthermore, we show that TD-VCL represents a spectrum of learning objectives that range from vanilla VCL to n-Step KL VCL. Finally, we present experiments on challenging versions of popular CL benchmarks, demonstrating that they outperform standard VCL and non-variational baselines (as shown in Figure 1), effectively alleviating Catastrophic Forgetting.

## 2 RELATED WORK

**Continual Learning** has been studied throughout the past decades, both in Artificial Intelligence (Schlimmer & Fisher, 1986; Sutton & Whitehead, 1993; Ring, 1997) and in Neuro- and Cognitive Sciences (Flesch et al., 2023; French, 1999; McCloskey & Cohen, 1989). More recently, the focus

has shifted towards overparameterized models, such as deep neural networks (Hadsell et al., 2020; Goodfellow et al., 2015; Serra et al., 2018; Adel et al., 2020). Given their powerful predictive capabilities, recent literature approaches CL from a wide range of perspectives. For instance, by regularizing the optimization objective to account for old tasks (Kirkpatrick et al., 2016; Zenke et al., 2017; Chaudhry et al., 2018); by replaying an external memory composed by a set of previous tasks (Lopez-Paz & Ranzato, 2017; Bang et al., 2021; Rebuffi et al., 2016); or by modifying the optimization procedure or manipulating the estimated gradients (Zeng et al., 2018; Javed & White, 2019; Liu & Liu, 2022). We refer to (Wang et al., 2024) for an extensive review of recent approaches. The proposed method in this work is placed between regularization-based and replay-based methods.

**Bayesian CL.** In the Bayesian framework, prior methods exploit the recursive relationship between subsequent posteriors that emerge from the Bayes' rule in the CL setting (Section 3). Since Bayesian inference is often intractable, they fundamentally differ in the design of approximated inference. We highlight works that learn posteriors via Laplace approximation (Ritter et al., 2018; Schwarz et al., 2018), sequential Bayesian Inference (Titsias et al., 2020; Pan et al., 2020), and Variational Inference (VI) (Nguyen et al., 2018; Loo et al., 2021). Our method lies in the latter category.

**Variational Inference for CL.** Variational Continual Learning (VCL) (Nguyen et al., 2018) introduced the idea of online VI for the Continual Learning setting. It leverages the Bayesian recursion of posteriors to build an optimization target for the next step's posterior based on the current one. Similarly, our work also optimizes a target based on previous approximated posteriors. On the other hand, rather than relying on a single past posterior estimation, it bootstraps on several previous estimations to prevent compounded errors. Nguyen et al. (2018) further incorporate an external replay buffer to prevent forgetting, requiring a two-step optimization. In contrast, our work only requires a single-step optimization as the replay mechanism naturally emerges from the learning objective.

Other derivative works usually blend VCL with architectural and optimization improvements (Loo et al., 2020; 2021; Guimeng et al., 2022; Tseran, 2018) or different posterior modeling assumptions (Auddy et al., 2020; Yang et al., 2019). We specifically highlight (Loo et al., 2021), which, among other contributions, introduces explicitly the likelihood-tempering hyperparameter, which is implicitly used in vanilla VCL and also in our work to address variational over-pruning (Trippe & Turner, 2018). Otherwise, the proposed innovations are orthogonal to this work.

# 3 PRELIMINARIES

**Problem Statement**. In the Continual Learning setting, a model learns from a streaming of tasks, which forms a non-stationary data distribution throughout time. More formally, we consider a task distribution $\mathcal{T}$ and represent each task $t \sim \mathcal{T}$ as a set of pairs $\{(\boldsymbol{x}_t, y_t)\}^{N_t}$, where $N_t$ is the dataset size. At every timestep t[1], the model receives a batch of data $\mathcal{D}_t$ for training. We evaluate the model in held-out test sets, considering all previously observed tasks.

In the **Bayesian framework** for CL, we assume a prior distribution over parameters $p(\boldsymbol{\theta})$, and the goal is to learn a posterior distribution $p(\boldsymbol{\theta} \mid \mathcal{D}_{1:T})$ after observing $T$ tasks. Crucially, given the sequential nature of tasks, we identify a recursive property of posteriors:

$$p(\boldsymbol{\theta} \mid \mathcal{D}_{1:T}) \propto p(\boldsymbol{\theta})p(\mathcal{D}_{1:T} \mid \boldsymbol{\theta}) \overset{\text{i.i.d}}{=} p(\boldsymbol{\theta}) \prod_{t=1}^{T} p(\mathcal{D}_t \mid \boldsymbol{\theta}) \propto p(\boldsymbol{\theta} \mid \mathcal{D}_{1:T-1})p(\mathcal{D}_T \mid \boldsymbol{\theta}), \quad (1)$$

where we assume that tasks are i.i.d. Equation 1 shows that we may update the posterior estimation online, given the likelihood of the subsequent task.

**Variational Continual Learning**. Despite the elegant recursion, computing the posterior $p(\boldsymbol{\theta} \mid \mathcal{D}_{1:T})$ exactly is often intractable, especially for large parameter spaces. Hence, we rely on an approximation. VCL achieves this by employing online variational inference (Ghahramani & Attias, 2000). It assumes the existence of variational parameters $q(\boldsymbol{\theta})$ whose goal is to approximate the posterior by minimizing the following KL divergence over a space of variational approximations $\mathcal{Q}$:

---

[1]We represent each task with the index $t$, which also denotes the timestep in the sequence of tasks.

$$q_t(\boldsymbol{\theta}) = \underset{q \in \mathcal{Q}}{\arg\min} \, \mathscr{D}_{KL}(q(\boldsymbol{\theta}) \, \| \, \frac{1}{Z_t} q_{t-1}(\boldsymbol{\theta}) p(\mathcal{D}_t \mid \boldsymbol{\theta})), \tag{2}$$

where $Z_t$ represents a normalization constant. The objective in Equation 2 is equivalent to maximizing the variational lower bound of the online marginal likelihood:

$$\mathcal{L}_{VCL}^t(\boldsymbol{\theta}) = \mathbb{E}_{\boldsymbol{\theta} \sim q_t(\boldsymbol{\theta})}[\log p(\mathcal{D}_t \mid \boldsymbol{\theta})] - \mathscr{D}_{KL}(q_t(\boldsymbol{\theta}) \, \| \, q_{t-1}(\boldsymbol{\theta})). \tag{3}$$

We can interpret the loss in Equation 3 through the lens of the stability-plasticity dilemma (Abraham & Robins, 2005). The first term maximizes the likelihood of the new task (encouraging plasticity), whereas the KL term penalizes parametrizations that deviate too far from the previous posterior estimation, which supposedly contains the knowledge from past tasks (encouraging memory stability).

## 4 TEMPORAL-DIFFERENCE VARIATIONAL CONTINUAL LEARNING

Maximizing the objective in Equation 3 is equivalent to the optimization in Equation 2, but its computation relies on two main approximations. First, computing the expected log-likelihood term analytically is not tractable, which requires a Monte-Carlo (MC) approximation. Second, the KL term relies on a previous posterior estimate, which may be biased from previous approximation errors. While updating the posterior to account for the next task, these biases deviate the learning target from the true objective. Crucially, as Equation 3 solely relies on the very latest posterior estimation, the error compounds with successive recursive updates.

Alternatively, we may represent the same objective as a function of several previous posterior estimations and alleviate the effect of the approximation error from any particular one. By considering several past estimates, the objective dilutes individual errors, allows correct posterior approximates to exert a corrective influence, and leverages a broader global context to the learning target, reducing the impact of compounding errors over time.

### 4.1 VARIATIONAL CONTINUAL LEARNING WITH n-STEP KL REGULARIZATION

We start by presenting a new objective that is equivalent to Equation 2 while also meeting the aforementioned desiderata:

**Proposition 4.1.** *The standard KL minimization objective in Variational Continual Learning (Equation 2) is equivalently represented as the following objective, where $n \in \mathbb{N}_0$ is a hyperparameter:*

$$q_t(\boldsymbol{\theta}) = \underset{q \in \mathcal{Q}}{\arg\max} \, \mathbb{E}_{\boldsymbol{\theta} \sim q_t(\boldsymbol{\theta})} \Big[ \sum_{i=0}^{n-1} \frac{(n-i)}{n} \log p(\mathcal{D}_{t-i} \mid \boldsymbol{\theta}) \Big] - \sum_{i=0}^{n-1} \frac{1}{n} \mathscr{D}_{KL}(q_t(\boldsymbol{\theta}) \, \| \, q_{t-i-1}(\boldsymbol{\theta})). \tag{4}$$

We present the proof of Proposition 4.1 in Appendix A. We name Equation 4 as the n-Step KL regularization objective. It represents the same learning target of Equation 2 as a sum of weighted likelihoods and KL terms that consider different posterior estimations, which can be interpreted as "distributing" the role of regularization among them. For instance, if an estimate $q_{t-i}$ deviates too far from the true posterior, it only affects $1/n$ of the KL regularization term. The hyperparameter $n$ assumes integer values up to $t$ and defines how far in the past the learning target goes. If $n$ is set to 1, we recover vanilla VCL.

An interesting insight comes from the likelihood term. It contains the likelihood of different tasks, weighted by their recency. Hence, the idea of re-training in old task data, commonly leveraged as a heuristic in CL methods, naturally emerges in the proposed objective. Additionally, we may estimate the likelihood term by replaying data from different tasks simultaneously, alleviating the violation of the i.i.d assumption that happens given the online, sequential nature of CL (Hadsell et al., 2020).

### 4.2 FROM n-STEP KL TO TEMPORAL-DIFFERENCE TARGETS

The learning objective in Equation 4 relies on several different posterior estimates, alleviating the compounding error problem. A caveat is that all estimates have the same weight in the final ob-

jective. One may want to have more flexibility by giving different weights for them – for instance, amplifying the effect from the most recent estimate while drastically reducing the impact of previous ones. It is possible to accomplish that, as shown in the following proposition:

**Proposition 4.2.** *The standard KL minimization objective in VCL (Equation 2) is equivalently represented as the following objective, with $n \in \mathbb{N}_0$, and $\lambda \in [0, 1)$ hyperparameters:*

$$\arg\max_{q \in \mathcal{Q}} \mathbb{E}_{\boldsymbol{\theta} \sim q_t(\boldsymbol{\theta})} \Big[ \sum_{i=0}^{n-1} \frac{\lambda^i (1 - \lambda^{n-i})}{1 - \lambda^n} \log p(\mathcal{D}_{t-i} \mid \boldsymbol{\theta}) \Big] - \sum_{i=0}^{n-1} \frac{\lambda^i (1 - \lambda)}{1 - \lambda^n} \mathscr{D}_{KL}(q_t(\boldsymbol{\theta}) \,\|\, q_{t-i-1}(\boldsymbol{\theta})). \tag{5}$$

The proof is available in Appendix B. We call Equation 5 the TD($\lambda$)-VCL objective[2]. It augments the n-Step KL Regularization to weight the regularization effect of different estimates in a way that geometrically decays – via the $\lambda^i$ term – as far as it goes in the past. Other $\lambda$-related terms serve as normalization constants. Equation 5 provides a more granular level of target control.

Interestingly, this objective relates intrinsically to the $\lambda$-returns for Temporal-Difference (TD) learning in valued-based reinforcement learning (Sutton & Barto, 2018). More broadly, both objectives of Equations 4 and 5 are compound updates that combine $n$-step Temporal-Difference targets, as shown below. First, we formally define a TD target in the CL context:

**Definition 4.3.** For a timestep $t$, the n-Step Temporal-Difference target for Variational Continual Learning is defined as, $\forall n \in \mathbb{N}_0$, $n \leq t$:

$$\text{TD}_t(n) = \mathbb{E}_{\boldsymbol{\theta} \sim q_t(\boldsymbol{\theta})} \left[ \sum_{i=0}^{n-1} \log p(\mathcal{D}_{t-i} \mid \boldsymbol{\theta}) \right] - \mathscr{D}_{KL}(q_t(\boldsymbol{\theta}) \,\|\, q_{t-n}(\boldsymbol{\theta})). \tag{6}$$

In Appendix C, we reveal the connection between Equation 6 and the TD targets employed in Reinforcement Learning, justifying the adopted terminology. From this definition, it follows that:

**Proposition 4.4.** $\forall n \in \mathbb{N}_0$, $n \leq t$ , *the objective in Equation 2 can be equivalently represented as:*

$$q_t(\boldsymbol{\theta}) = \arg\max_{q \in \mathcal{Q}} \text{TD}_t(n), \tag{7}$$

*with $\text{TD}_t(n)$ as in Definition 4.3. Furthermore, the objective in Equation 5 can also be represented as:*

$$q_t(\boldsymbol{\theta}) = \arg\max_{q \in \mathcal{Q}} \frac{1 - \lambda}{1 - \lambda^n} \underbrace{\left[ \sum_{k=0}^{n-1} \lambda^k \text{TD}_t(k+1) \right]}_{\text{Discounted sum of TD targets}}. \tag{8}$$

The proof is in Appendix D. Proposition 4.4 states that the TD($\lambda$)-VCL objective is a sum of discounted TD targets (up to a normalization constant), effectively representing $\lambda$-returns. Parallelly, one can show that the n-Step KL Regularization objective, as a particular case, is a simple average of n-Step TD targets. Fundamentally, the key idea behind these objectives is *bootstrapping*: they build a learning target estimate based on other estimates. Ultimately, the "$\lambda$-target" in Equation 5 provides flexibility for bootstrapping by allowing multiple previous estimates to influence the objective.

**The TD-VCL objectives generalize a spectrum of Continual Learning algorithms**. As a final remark, in Appendix E, we show that, based on the choice of hyperparameters, the TD($\lambda$)-VCL objective forms a family of learning algorithms that span from Vanilla VCL to n-Step KL Regularization. Fundamentally, it mixes different targets of MC approximations for expected log-likelihood and KL regularization. This process is similar to how TD($\lambda$) and $n$-step TD mix MC updates and TD predictions in Reinforcement Learning, effectively providing a mechanism to strike a balance between the variance from MC estimations and the bias from bootstrapping (Sutton & Barto, 2018).

---

[2]We refer to both n-Step KL Regularization and TD($\lambda$)-VCL as TD-VCL objectives.

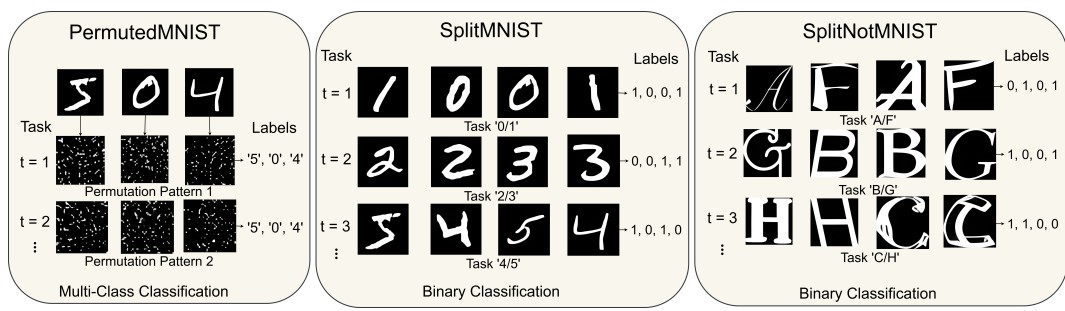

Figure 3: **The PermutedMNIST, SplitMNIST, and SplitNotMNIST benchmarks**. In the PermutedMNIST (left), each task is a different permutation pattern of pixels from MNIST. In SplitMNIST (middle), each task is a binary classification problem between two digits. SplitNotMNIST (right) is similar to SplitMNIST but comprises a harder dataset of characters with diverse font styles.

## 5 EXPERIMENTS AND DISCUSSION

Our central hypothesis is that leveraging multiple past posterior estimates mitigates the impact of compounded errors inherent to the VCL objective, thus alleviating the problem of Catastrophic Forgetting. We now provide an experimental setup for validation. Specifically, we evaluate this hypothesis by analyzing the three questions highlighted in Section 5.2.

**Implementation**. We use a Gaussian mean-field approximate posterior and assume a Gaussian prior $\mathcal{N}(0, \sigma^2 I)$, and parameterize all distributions as deep networks. For all variational objectives, we compute the KL term analytically and employ Monte Carlo approximations for the expected log-likelihood terms, leveraging the reparametrization trick (Kingma & Welling, 2014) for computing gradients. We employed likelihood-tempering (Loo et al., 2021) to prevent variational over-pruning (Trippe & Turner, 2018). Lastly, for test-time evaluation, we compute the posterior predicting distribution by marginalizing out the approximated posterior distribution via Monte-Carlo sampling. We provide further details in Appendix F and our code[3].

**Baselines**. We compare TD-VCL and n-Step KL VCL with several baselines. **Online MLE** naively applies maximum likelihood estimation in the current task data. It serves as a lower bound for other methods, as well as a way to evaluate how challenging the benchmark is. **Batch MLE** applies maximum likelihood estimation considering a buffer of current and old task data. **VCL**, introduced by Nguyen et al. (2018), optimizes the objective in Equation 3. **VCL CoreSet** is a VCL variant that incorporates a replay set to mitigate any residual forgetting (Nguyen et al., 2018).

**Benchmarks**. We consider three Continual Learning benchmarks, illustrated in Figure 3. The **PermutedMNIST** (Goodfellow et al., 2015) is a multi-class classification setup where each task corresponds to a different permutation of pixels in the MNIST data. The benchmark runs ten successive tasks. The **SplitMNIST** (Zenke et al., 2017) is a binary classification setting where the model needs to recognize between pairs of digits. Five tasks arrive sequentially: 0/1, 2/3, 4/5, 6/7, 8/9. The challenging **SplitNotMNIST** (Nguyen et al., 2018) contains characters from diverse font styles, comprising 400,000 examples. Similar to SplitMNIST, it comprises five sequential tasks to recognize pairs of characters: A/F, B/G, C/H, D/I, and E/J. For all benchmarks, each evaluation iteration considers the performance in all past tasks, where we report averaged accuracy across them.

Before turning our attention to evaluating the proposed TD-VCL objective, our first empirical contribution is a critical evaluation of common design choices presented by previous methods while employing these benchmarks.

### 5.1 TOWARDS HIGHER STANDARDS FOR CONTINUAL LEARNING EVALUATION

Popular Continual Learning benchmarks (Goodfellow et al., 2015; Zenke et al., 2017; Nguyen et al., 2018) provide an effective experimental setup. These benchmarks offer tasks that, while conceptually simple in isolation, present a challenging task streaming setup that highlights the phenomenon of Catastrophic Forgetting. This combination facilitates the study of Continual Learning meth-

---

[3]https://anonymous.4open.science/r/vcl-nstepkl-5707

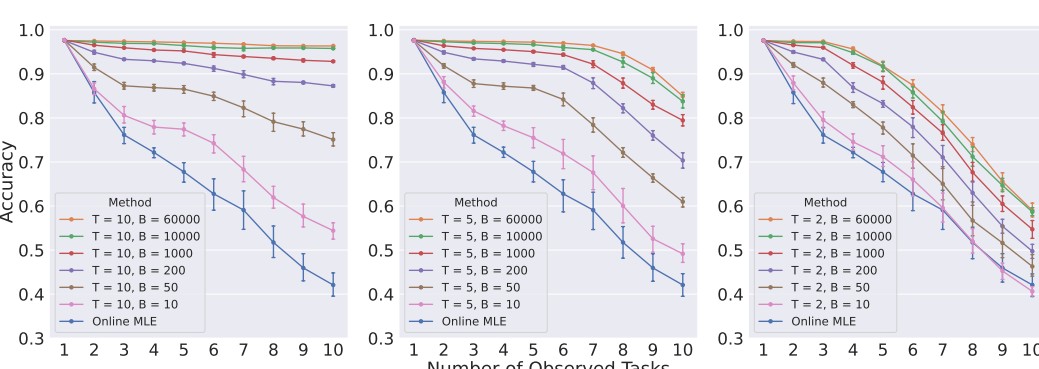

Figure 4: **A Replay Buffer analysis on the PermutedMNIST**. Each curve represents a model re-trained on a buffer composed of "$T$" previous tasks, "$B$" examples of each. Online MLE only considers the current task. Allowing "unlimited" access to previous task data trivializes the CL setting, and a simple MLE baseline is enough to attain strong results. Nevertheless, as we restrict the replay buffer in size and number of tasks, the benchmark becomes substantially more challenging and shows signs of Catastrophic Forgetting.

ods through rapid iterations and modest deep architectures, making it ideal for academic settings. Nonetheless, we argue that the "unrestricted" versions of these benchmarks – specifically regarding replay buffer adoption and model architecture – are either trivially addressed by simple baselines or do not reflect realistic scenarios of Continual Learning. This observation motivates our work to incorporate certain restrictions in the considered methods, resulting in a more challenging setup for Continual Learning while maintaining the benchmarks' original desiderata.

**"Unlimited" training on old task data trivializes CL benchmarks**. Figure 4 presents MLE models trained on different levels of old task data (besides the data from the current task). Online MLE means no usage of data from old tasks. On the flip side, we re-train the remaining models considering the data of $T$ previous tasks, with $B$ examples of each. It shows that allowing access to all the old tasks is enough for an MLE model to maintain high accuracy even when presenting to only a set as tiny as 200 examples. As we reduce the number of old tasks in the buffer, performance decreases, showing clear signs of Catastrophic Forgetting. For $T = 2$, all models present an accuracy lower than 60% regardless of the volume of old task data. Therefore, in order to avoid trivializing the benchmark, we impose additional restrictions for re-training in prior tasks. For PermutedMNIST, we restrict re-training to the two most recent past tasks, with 200 examples per task; for others, we allow only the most recent past task with 40 examples. As shown in Figure 4, MLE-based methods do not perform well in this setting.

**"Multi-Head" Classifiers are unrealistic for CL**. "Multi-Head" networks train a different classifier for each task on top of a shared backbone. As argued by Farquhar & Gal (2018), it is not well suited for CL for two reasons: first, it assumes *a priori* knowledge about the number of tasks, which is a strong assumption that simplifies the problem to a multi-task learning setup. Second, it provides independent parameters for each task, which naturally circumvents Catastrophic Forgetting by disregarding the effect of negative transfer among tasks. In Appendix H, we evaluate the methods on the SplitMNIST benchmark considering the multi-head and single-head classifiers. In the former, all baselines trivially attain high average accuracy; in the latter, all methods face a much more challenging setup. Hence, we adopt single-head architecture throughout this work.

Lastly, we highlight that all evaluated methods – including the proposed ones – are subject to the adopted restrictions highlighted in this Section. Therefore, they are trained in the same data with the same parametrization, ensuring a fair comparison setup.

## 5.2 EXPERIMENTS

We highlight and analyze the following questions to evaluate our hypothesis and proposed method:

Table 1: **Quantitative comparison on the PermutedMNIST, SplitMNIST, and SplitNotMNIST benchmarks**. Each column presents the average accuracy across the past $t$ observed tasks. Results are reported with two standard deviations across ten seeds. Top two results are in **bold**, while noticeably lower results are in gray. TD-VCL objective consistently outperforms standard VCL variants, especially when the number of observed tasks increase.

| | PermutedMNIST | | | | | | | | |
| | $t=2$ | $t=3$ | $t=4$ | $t=5$ | $t=6$ | $t=7$ | $t=8$ | $t=9$ | $t=10$ |
|---|---|---|---|---|---|---|---|---|---|
| Online MLE | 0.87±0.07 | 0.77±0.06 | 0.73±0.08 | 0.69±0.08 | 0.65±0.13 | 0.57±0.16 | 0.51±0.14 | 0.46±0.11 | 0.40±0.08 |
| Batch MLE | 0.95±0.01 | 0.93±0.01 | 0.88±0.04 | 0.83±0.04 | 0.77±0.10 | 0.71±0.13 | 0.64±0.12 | 0.57±0.11 | 0.51±0.06 |
| VCL | 0.95±0.00 | 0.94±0.01 | 0.93±0.02 | 0.91±0.02 | 0.89±0.03 | 0.86±0.03 | 0.83±0.04 | 0.80±0.06 | 0.78±0.04 |
| VCL CoreSet | **0.96±0.00** | **0.95±0.00** | **0.94±0.00** | **0.93±0.02** | 0.91±0.01 | 0.89±0.02 | 0.86±0.03 | 0.84±0.04 | 0.81±0.03 |
| n-Step TD-VCL | 0.95±0.01 | 0.94±0.00 | **0.94±0.00** | **0.93±0.01** | **0.92±0.01** | **0.91±0.01** | **0.90±0.02** | **0.89±0.01** | **0.88±0.02** |
| TD($\lambda$)-VCL | **0.97±0.00** | **0.96±0.00** | **0.95±0.00** | **0.94±0.01** | **0.93±0.01** | **0.92±0.01** | **0.91±0.01** | **0.90±0.01** | **0.89±0.02** |

| | Split MNIST | | | | | Split NotMNIST | | | |
| | $t=2$ | $t=3$ | $t=4$ | $t=5$ | | $t=2$ | $t=3$ | $t=4$ | $t=5$ |
|---|---|---|---|---|---|---|---|---|---|
| Online MLE | 0.86±0.02 | 0.61±0.03 | 0.75±0.04 | 0.57±0.06 | | 0.72±0.02 | 0.61±0.05 | 0.61±0.00 | 0.51±0.04 |
| Batch MLE | 0.95±0.04 | 0.65±0.04 | 0.82±0.04 | 0.59±0.03 | | 0.71±0.02 | 0.65±0.03 | 0.61±0.00 | 0.50±0.06 |
| VCL | 0.87±0.02 | 0.66±0.04 | 0.82±0.03 | 0.64±0.11 | | 0.69±0.04 | 0.63±0.03 | 0.60±0.00 | 0.51±0.06 |
| VCL CoreSet | 0.93±0.04 | 0.68±0.07 | 0.84±0.04 | 0.62±0.03 | | 0.69±0.04 | 0.65±0.02 | 0.60±0.01 | 0.51±0.07 |
| n-Step TD-VCL | **0.98±0.01** | **0.79±0.08** | **0.88±0.04** | **0.67±0.04** | | **0.72±0.04** | **0.73±0.05** | **0.70±0.04** | **0.58±0.08** |
| TD($\lambda$)-VCL | **0.98±0.01** | **0.81±0.07** | **0.89±0.03** | **0.66±0.02** | | **0.74±0.02** | **0.73±0.03** | **0.69±0.03** | **0.58±0.09** |

**Do the TD-VCL objectives effectively alleviate Catastrophic Forgetting in challenging CL benchmarks?** Table 1 presents the results in the three benchmarks, adopting the restrictions in Section 5.1 for all considered methods. Each column presents the average accuracy across the past $t$ observed tasks, and we present the results starting from $t = 2$ since $t = 1$ is simply single-task learning. For **PermutedMNIST** (top), all methods presented high accuracy for $t = 2$, suggesting that they could fit the data successfully. As the number of tasks increases, all methods start manifesting Catastrophic Forgetting at different levels. While Online and Batch MLE drastically suffer, variational approaches considerably retain old tasks' performance. The Core Set slightly helps VCL, and both n-Step KL and TD-VCL outperform them by a considerable margin, attaining approximately 90% average accuracy

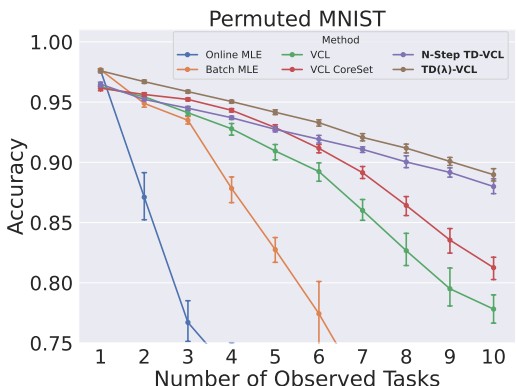

Figure 5: **Average accuracy across all observed tasks in PermutedMNIST**. The TD-VCL objectives lead to a substantial improvement against standard VCL and non-variational approaches.

after all tasks. For completeness, Figure 5 graphically shows the results. We emphasize the discrepancy between variational approaches and naive baselines and highlight the performance boost by adopting TD-VCL objectives.

For **SplitMNIST** (bottom-left), we highlight that the TD-VCL objectives also surpassed baselines in all configurations. Nonetheless, all methods present a significant decrease in performance for $t = 5$, suggesting a more challenging setup for addressing Catastrophic Forgetting that opens a venue for future research. We discuss SplitMNIST results in more detail in Appendix H.

Lastly, the **SplitNotMNIST** (bottom-right) is a considerably harder benchmark, as the letters come from a diverse set of font styles. Furthermore, we purposely decided to employ a modest network architecture (as for previous benchmarks). Facing hard tasks with less expressive parametrizations will result in higher posterior approximation error. Our goal is to evaluate how the variational methods behave in this setting. Naturally, all methods struggle to fit the tasks. For PermutedMNIST and SplitMNIST, all models could at least fit the current task almost perfectly, while presenting accura-

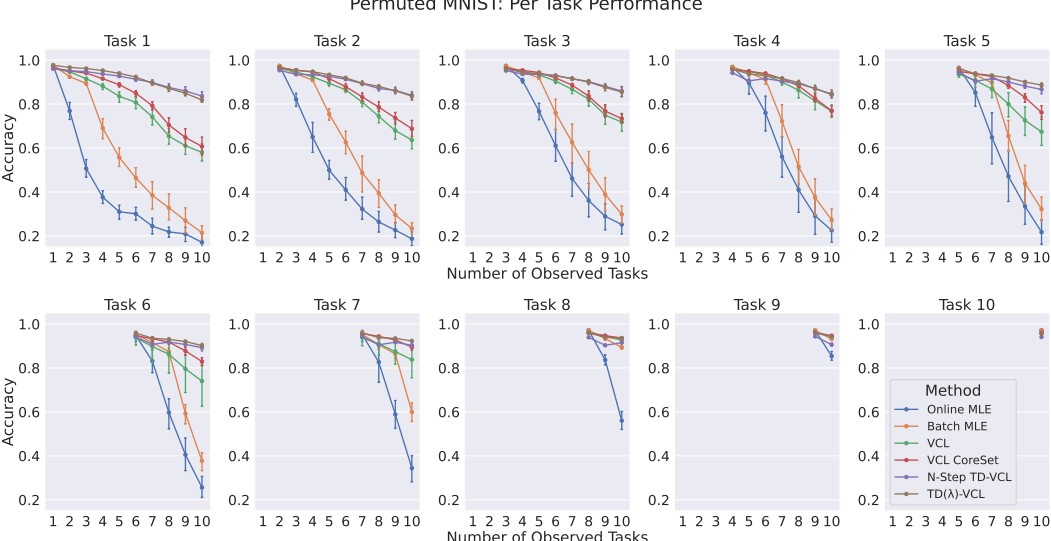

Figure 6: **Per-task performance (accuracy) over time in the PermutedMNIST benchmark**. Each plot represents the accuracy of one task (identified in the plot title) while the number of observed tasks increases. We highlight a stronger effect of Catastrophic Forgetting on earlier tasks for the baselines, while TD-VCL objectives are noticeably more robust to this phenomenon.

cies around 80% and 90% for current tasks in SplitNotMNIST, as shown in Figure 9 in Appendix I. This validates our intent in this setting.

Once again, n-step KL and TD-VCL surpassed the baselines after observing more than three tasks. The effect is more pronounced after increasing the number of observed tasks. These objectives are the only ones whose resultant models achieved non-trivial average accuracy after observing all tasks. This evidence suggests that leveraging multiple posterior estimates during learning is better than only the latest one, even where the approximation error is high.

**How do the TD-VCL objectives affect per-task performance?** While Table 1 presents performance averaged across different tasks, we now analyze the accuracy of each task separately in the course of online learning. This setup is relevant since solely considering the averaged accuracy may hide a stronger Catastrophic Forgetting effect from earlier tasks by "compensating" with higher accuracy from later tasks. We show the results for PermutedMNIST in Figure 6, while we defer the results for SplitMNIST and SplitNotMMNIST for Appendices H and I, respectively. Figure 6 presents a sequence of plots, where each figure represents the accuracy of one task while the number of observed tasks increases. Naturally, the tasks that appear at later stages present fewer data points: for instance, "Task 10" has a single data point as it does not have test data for earlier timesteps.

As observed, per-task performance explicitly shows a stronger effect of Catastrophic Forgetting for earlier tasks in the adopted baselines. We particularly highlight how non-variational approaches fail for them. In this direction, TD-VCL objectives presented a more robust performance against others. For instance, we highlight the results for Task 1. After observing all tasks, the proposed methods demonstrated accuracy of around 80% and 85%. The VCL baselines dropped to 50% and 60%, and MLE-based methods failed with only 20% of accuracy.

**How do the TD-VCL objectives behave with the choice of the hyperparameters $n$, $\lambda$, and the likelihood-tempering parameter $\beta$?** The proposed learning objectives introduce two new hyperparameters: $n$ (the number of considered previous posterior estimates in the learning target) and $\lambda$ for TD($\lambda$)-VCL (which controls the level of influence for each past posterior estimate). Furthermore, it also inherits the $\beta$ parameter from the standard VCL. Hence, we evaluate the sensitivity of the proposed objectives concerning these hyperparameters, presenting results and detailed discussion in Appendix J. We highlight three main findings. First, similarly to VCL, TD-VCL objectives are sensitive to the likelihood-tempering hyperparameter. Second, increasing $n$ is beneficial up to a certain point, from which it becomes detrimental, suggesting that leveraging too old posterior estimates

may not be useful. Lastly, TD-VCL objectives present robustness over the choice of $\lambda$, with a more pronounced effect when the number of observed tasks increases.

## 6 CLOSING REMARKS

In this work, we have presented a new family of variational objectives for Continual Learning, namely Temporal-Difference VCL. TD-VCL is an unbiased proxy of the standard VCL objective but leverages several previous posterior estimates to alleviate the compounding error caused by recursive approximations. We have shown that TD-VCL represents a spectrum of continual learning algorithms and is equivalent to a discounted sum of n-step Temporal-Difference targets. Lastly, we have empirically displayed that it helps address Catastrophic Forgetting, surpassing vanilla VCL and other baselines in improved versions of popular CL benchmarks.

**Limitations**. Despite being theoretically principled and attaining superior performance, TD-VCL presents limitations. First, the hyperparameters $n$ and $\lambda$ depend on the evaluated setting, which may require certain tuning. Second, the objectives require maintaining a copy of the past $n$ posterior estimates, increasing the memory requirements. Still, we believe this is not a major limitation as TD-VCL suits well modern deep Bayesian architectures that target smaller parameter subspaces for posterior approximation (Yang et al., 2024; Dwaracherla et al., 2024; Melo et al., 2024).

**Future Work**. While presenting connections with Temporal-Difference methods, our work does not claim that TD-VCL is an RL algorithm. Further mathematical connections with Markov Decision/Reward Processes formalism are left as future work. Another interesting direction is to apply TD-VCL objectives for probabilistic meta-learning (Finn et al., 2018; Zintgraf et al., 2020).

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

## A  DERIVATION OF THE n-STEP KL REGULARIZATION OBJECTIVE

In this Section, we prove Proposition 4.1:

**Proposition 4.1.** *The standard KL minimization objective in Variational Continual Learning (Equation 2) is equivalently represented as the following objective, where $n \in \mathbb{N}_0$ is a hyperparameter:*

$$q_t(\boldsymbol{\theta}) = \arg\max_{q \in \mathcal{Q}} \mathbb{E}_{\boldsymbol{\theta} \sim q_t(\boldsymbol{\theta})} \Big[ \sum_{i=0}^{n-1} \frac{(n-i)}{n} \log p(\mathcal{D}_{t-i} \mid \boldsymbol{\theta}) \Big] - \sum_{i=0}^{n-1} \frac{1}{n} \mathscr{D}_{KL}(q_t(\boldsymbol{\theta}) \mid\mid q_{t-i-1}(\boldsymbol{\theta})). \quad (4)$$

*Proof.* Starting from Equation 2, we can expand it as a sum of equal terms and utilize the recursive property (Equation 1) to expand these terms:

$$q_t(\boldsymbol{\theta}) = \arg\min_{q \in \mathcal{Q}} \mathscr{D}_{KL}(q(\boldsymbol{\theta}) \mid\mid \frac{1}{Z_t} q_{t-1}(\boldsymbol{\theta}) p(\mathcal{D}_t \mid \boldsymbol{\theta}))$$

$$= \arg\min_{q \in \mathcal{Q}} \frac{n}{n} \mathscr{D}_{KL}(q(\boldsymbol{\theta}) \mid\mid \frac{1}{Z_t} q_{t-1}(\boldsymbol{\theta}) p(\mathcal{D}_t \mid \boldsymbol{\theta}))$$

$$= \arg\min_{q \in \mathcal{Q}} \frac{1}{n} \Bigg[ \mathscr{D}_{KL}(q(\boldsymbol{\theta}) \mid\mid \frac{1}{Z_t} q_{t-1}(\boldsymbol{\theta}) p(\mathcal{D}_t \mid \boldsymbol{\theta}))$$

$$+ \mathscr{D}_{KL}(q(\boldsymbol{\theta}) \mid\mid \frac{1}{Z_t Z_{t-1}} q_{t-2}(\boldsymbol{\theta}) p(\mathcal{D}_t \mid \boldsymbol{\theta}) p(\mathcal{D}_{t-1} \mid \boldsymbol{\theta})) + \dots$$

$$+ \mathscr{D}_{KL}(q(\boldsymbol{\theta}) \mid\mid \frac{1}{\prod_{i=0}^{n-1} Z_{t-i}} q_{t-n}(\boldsymbol{\theta}) \prod_{i=0}^{n-1} p(\mathcal{D}_{t-i} \mid \boldsymbol{\theta})) \Bigg]$$

$$= \arg\min_{q \in \mathcal{Q}} \frac{1}{n} \Bigg[ \mathscr{D}_{KL}(q_t(\boldsymbol{\theta}) \mid\mid q_{t-1}(\boldsymbol{\theta})) - \mathbb{E}_{\boldsymbol{\theta} \sim q_t(\boldsymbol{\theta})}[\log p(\mathcal{D}_t \mid \boldsymbol{\theta})]$$

$$+ \mathscr{D}_{KL}(q_t(\boldsymbol{\theta}) \mid\mid q_{t-2}(\boldsymbol{\theta})) - \mathbb{E}_{\boldsymbol{\theta} \sim q_t(\boldsymbol{\theta})}[\log p(\mathcal{D}_t \mid \boldsymbol{\theta}) + \log p(\mathcal{D}_{t-1} \mid \boldsymbol{\theta})] + \dots$$

$$+ \mathscr{D}_{KL}(q_t(\boldsymbol{\theta}) \mid\mid q_{t-n}(\boldsymbol{\theta})) - \mathbb{E}_{\boldsymbol{\theta} \sim q_t(\boldsymbol{\theta})}[\sum_{i=0}^{n-1} \log p(\mathcal{D}_{t-i} \mid \boldsymbol{\theta})] \Bigg]$$

$$= \arg\min_{q \in \mathcal{Q}} \frac{1}{n} \Bigg[ \sum_{i=0}^{n-1} \mathscr{D}_{KL}(q_t(\boldsymbol{\theta}) \mid\mid q_{t-i}(\boldsymbol{\theta})) - \mathbb{E}_{\boldsymbol{\theta} \sim q_t(\boldsymbol{\theta})} \Big[ n \log p(\mathcal{D}_t \mid \boldsymbol{\theta})$$

$$+ (n-1) \log p(\mathcal{D}_{t-1} \mid \boldsymbol{\theta}) + \dots + \log p(\mathcal{D}_{t-n+1} \mid \boldsymbol{\theta}) \Big] \Bigg]$$

$$= \arg\max_{q \in \mathcal{Q}} \mathbb{E}_{\boldsymbol{\theta} \sim q_t(\boldsymbol{\theta})} \Big[ \sum_{i=0}^{n-1} \frac{(n-i)}{n} \log p(\mathcal{D}_{t-i} \mid \boldsymbol{\theta}) \Big] - \sum_{i=0}^{n-1} \frac{1}{n} \mathscr{D}_{KL}(q_t(\boldsymbol{\theta}) \mid\mid q_{t-i-1}(\boldsymbol{\theta})).$$

$$(9)$$

$\square$

## B  DERIVATION OF THE TEMPORAL-DIFFERENCE VCL OBJECTIVE

Before proving Proposition 4.2, we start by presenting a well known result for the sum of geometric series:

**Lemma B.1.** *The finite sum of a geometric series with $n$ terms, common ratio $\lambda$ and initial term $a$ is given by:*

$$\sum_{k=0}^{n-1} \lambda^k a = \frac{a(1 - \lambda^n)}{(1 - \lambda)} \tag{10}$$

*Proof.* Let $s_n = \sum_{k=0}^{n} \lambda^k a$. Hence,

$$\begin{aligned}
s_n - \lambda s_n &= \sum_{k=0}^{n-1} \lambda^k a - \lambda \sum_{k=0}^{n-1} \lambda^k a = a - a\lambda^n \\
&\iff s_n(1 - \lambda) = a(1 - \lambda^n) \\
&\iff s_n = \frac{a(1 - \lambda^n)}{(1 - \lambda)}.
\end{aligned} \tag{11}$$

$\square$

Now, we prove Proposition 4.2.

**Proposition 4.2.** *The standard KL minimization objective in VCL (Equation 2) is equivalently represented as the following objective, with $n \in \mathbb{N}_0$, and $\lambda \in [0, 1)$ hyperparameters:*

$$\arg\max_{q \in \mathcal{Q}} \mathbb{E}_{\boldsymbol{\theta} \sim q_t(\boldsymbol{\theta})} \Big[ \sum_{i=0}^{n-1} \frac{\lambda^i(1 - \lambda^{n-i})}{1 - \lambda^n} \log p(\mathcal{D}_{t-i} \mid \boldsymbol{\theta}) \Big] - \sum_{i=0}^{n-1} \frac{\lambda^i(1 - \lambda)}{1 - \lambda^n} \mathscr{D}_{KL}(q_t(\boldsymbol{\theta}) \mid\mid q_{t-i-1}(\boldsymbol{\theta})).$$

$$\tag{5}$$

*Proof.* We can use Lemma B.1 to expand the sum of KL terms:

$$
\begin{aligned}
q_t(\boldsymbol{\theta}) &= \underset{q \in \mathcal{Q}}{\arg\min}\ \mathscr{D}_{KL}(q(\boldsymbol{\theta}) \,\|\, \tfrac{1}{Z_t} q_{t-1}(\boldsymbol{\theta}) p(\mathcal{D}_t \mid \boldsymbol{\theta})) \\
&= \underset{q \in \mathcal{Q}}{\arg\min}\ \frac{1-\lambda}{1-\lambda^n} \frac{1-\lambda^n}{1-\lambda} \mathscr{D}_{KL}(q(\boldsymbol{\theta}) \,\|\, \tfrac{1}{Z_t} q_{t-1}(\boldsymbol{\theta}) p(\mathcal{D}_t \mid \boldsymbol{\theta})) \\
&= \underset{q \in \mathcal{Q}}{\arg\min}\ \frac{1-\lambda}{1-\lambda^n} \Bigg[ \mathscr{D}_{KL}(q(\boldsymbol{\theta}) \,\|\, \tfrac{1}{Z_t} q_{t-1}(\boldsymbol{\theta}) p(\mathcal{D}_t \mid \boldsymbol{\theta})) \\
&\qquad\qquad + \lambda \mathscr{D}_{KL}(q(\boldsymbol{\theta}) \,\|\, \tfrac{1}{Z_t Z_{t-1}} q_{t-2}(\boldsymbol{\theta}) p(\mathcal{D}_t \mid \boldsymbol{\theta}) p(\mathcal{D}_{t-1} \mid \boldsymbol{\theta})) + \ldots \\
&\qquad\qquad + \lambda^{n-1} \mathscr{D}_{KL}(q(\boldsymbol{\theta}) \,\|\, \tfrac{1}{\prod_{i=0}^{n-1} Z_{t-i}} q_{t-i}(\boldsymbol{\theta}) \prod_{i=0}^{n-1} p(\mathcal{D}_{t-i} \mid \boldsymbol{\theta})) \Bigg] \\
&= \underset{q \in \mathcal{Q}}{\arg\min}\ \frac{1-\lambda}{1-\lambda^n} \Bigg[ \mathscr{D}_{KL}(q_t(\boldsymbol{\theta}) \,\|\, q_{t-1}(\boldsymbol{\theta})) - \mathbb{E}_{\boldsymbol{\theta} \sim q_t(\boldsymbol{\theta})}[\log p(\mathcal{D}_t \mid \boldsymbol{\theta})] \\
&\qquad\qquad + \lambda \mathscr{D}_{KL}(q_t(\boldsymbol{\theta}) \,\|\, q_{t-2}(\boldsymbol{\theta})) - \lambda \mathbb{E}_{\boldsymbol{\theta} \sim q_t(\boldsymbol{\theta})}[\log p(\mathcal{D}_t \mid \boldsymbol{\theta}) + \log p(\mathcal{D}_{t-1} \mid \boldsymbol{\theta})] + \ldots \\
&\qquad\qquad + \lambda^{n-1} \mathscr{D}_{KL}(q_t(\boldsymbol{\theta}) \,\|\, q_{t-n}(\boldsymbol{\theta})) - \lambda^{n-1} \mathbb{E}_{\boldsymbol{\theta} \sim q_t(\boldsymbol{\theta})}\Big[\sum_{i=0}^{n-1} \log p(\mathcal{D}_{t-i} \mid \boldsymbol{\theta})\Big] \Bigg] \\
&= \underset{q \in \mathcal{Q}}{\arg\min}\ \frac{1-\lambda}{1-\lambda^n} \Bigg[ \sum_{i=0}^{n-1} \lambda^i \mathscr{D}_{KL}(q_t(\boldsymbol{\theta}) \,\|\, q_{t-i-1}(\boldsymbol{\theta})) - \mathbb{E}_{\boldsymbol{\theta} \sim q_t(\boldsymbol{\theta})}\Big[ \sum_{i=0}^{n-1} \lambda^i \log p(\mathcal{D}_t \mid \boldsymbol{\theta}) \\
&\qquad\qquad + \sum_{i=1}^{n-1} \lambda^i \log p(\mathcal{D}_{t-1} \mid \boldsymbol{\theta}) + \cdots + \lambda^{n-1} \log p(\mathcal{D}_{t-n+1} \mid \boldsymbol{\theta}) \Big] \Bigg] \\
&= \underset{q \in \mathcal{Q}}{\arg\min}\ \frac{1-\lambda}{1-\lambda^n} \Bigg[ \sum_{i=0}^{n-1} \lambda^i \mathscr{D}_{KL}(q_t(\boldsymbol{\theta}) \,\|\, q_{t-i-1}(\boldsymbol{\theta})) - \mathbb{E}_{\boldsymbol{\theta} \sim q_t(\boldsymbol{\theta})}\Big[ \frac{1-\lambda^n}{1-\lambda} \log p(\mathcal{D}_t \mid \boldsymbol{\theta}) \\
&\qquad\qquad + \frac{\lambda(1-\lambda^{n-1})}{1-\lambda} \log p(\mathcal{D}_{t-1} \mid \boldsymbol{\theta}) + \cdots + \lambda^{n-1} \log p(\mathcal{D}_{t-n+1} \mid \boldsymbol{\theta}) \Big] \Bigg] \\
&= \underset{q \in \mathcal{Q}}{\arg\max}\ \mathbb{E}_{\boldsymbol{\theta} \sim q_t(\boldsymbol{\theta})}\Big[ \sum_{i=0}^{n-1} \frac{\lambda^i(1-\lambda^{n-i})}{1-\lambda^n} \log p(\mathcal{D}_{t-i} \mid \boldsymbol{\theta}) \Big] - \sum_{i=0}^{n-1} \frac{\lambda^i(1-\lambda)}{1-\lambda^n} \mathscr{D}_{KL}(q_t(\boldsymbol{\theta}) \,\|\, q_{t-i-1}(\boldsymbol{\theta})).
\end{aligned}
$$

$$\tag{12}$$

$\square$

## C THE CONNECTION OF TD TARGETS IN TD-VCL AND REINFORCEMENT LEARNING

In the Section 4, we formalize the concept of n-Step Temporal-Difference for the Variational CL objective (Definition 4.3). In this Section, we reveal the connections between this definition and the widely used Temporal-Difference methods in Reinforcement Learning. Our aim is to clarify why Equation 6 indeed represents a temporal-difference target, both in a broad and strict senses.

In a **broad** sense, *bootstrapping* characterizes a Temporal-Difference target: building a learning target estimate based on previous estimates. Crucially, the leveraged estimates are functions of different timesteps. TD-VCL objectives applies bootstrapping in the KL regularization term, by considering one or more of posteriors estimates from previous timesteps.

In a **strict** sense, we can show that Equation 6 deeply resembles TD targets in Reinforcement Learning. RL assumes the formalism of a Markov Decision Process (MDP), defined by a tuple $\mathcal{M} = (\mathcal{S}, \mathcal{A}, \mathcal{P}, \mathcal{R}, \mathcal{P}_0, \gamma, H)$, where $\mathcal{S}$ is a state space, $\mathcal{A}$ is an action space, $\mathcal{P} : \mathcal{S} \times \mathcal{A} \times \mathcal{S} \to [0, \infty)$ is a transition dynamics, $\mathcal{R} : \mathcal{S} \times \mathcal{A} \to [-R_{max}, R_{max}]$ is a bounded reward function, $\mathcal{P}_0 : \mathcal{S} \to [0, \infty)$ is an initial state distribution, $\gamma \in [0, 1]$ is a discount factor, and $H$ is the horizon.

The standard RL objective is to find a policy that maximizes the cumulative reward:

$$\pi_{\boldsymbol{\theta}}^* = \arg\max_{\pi} \mathbb{E}_{\pi}[\sum_{k=0}^{H} \gamma^k \mathcal{R}(s_{t+k}, a_{t+k})], \tag{13}$$

with $a_t \sim \pi_{\boldsymbol{\theta}}(a_t \mid s_t)$, $s_t \sim \mathcal{P}(s_t \mid s_{t-1}, a_{t-1})$, and $s_0 \sim \mathcal{P}_0(s)$, where $\pi_{\boldsymbol{\theta}} : \mathcal{S} \times \mathcal{A} \to [0, \infty)$ is a policy parameterized by $\boldsymbol{\theta}$. Hence, we can define the following learning target, which represents a "value" function at each state $s_t$:

$$v_\pi(s_t) := \mathbb{E}_{\pi}[\sum_{k=0}^{H} \gamma^k \mathcal{R}(s_{t+k}, a_{t+k}) \mid s = s_t], \forall s_t \in \mathcal{S}. \tag{14}$$

Naturally, it follows that $\pi_{\boldsymbol{\theta}}^* = \arg\max_{\pi} v_\pi(s), \forall s \in \mathcal{S}$. Crucially, we can expand Equation 14 as follows:

$$
\begin{aligned}
v_\pi(s_t) &:= \mathbb{E}_{\pi}[\sum_{k=0}^{H} \gamma^k \mathcal{R}(s_{t+k}, a_{t+k}) \mid s = s_t] \\
&= \mathbb{E}_{\pi}[\mathcal{R}(s_t, a_t) + \sum_{k=1}^{H} \gamma^k \mathcal{R}(s_{t+k}, a_{t+k}) \mid s = s_t] \\
&= \mathbb{E}_{\pi}[\mathcal{R}(s_t, a_t) + \gamma v_\pi(s_{t+1})], \\
&= \mathbb{E}_{\pi}[\mathcal{R}(s_t, a_t) + \gamma \mathcal{R}(s_{t+1}, a_{t+1}) + \gamma^2 v_\pi(s_{t+2})], \\
&= \mathbb{E}_{\pi}[\sum_{k=0}^{n-1} \gamma^k \mathcal{R}(s_t, a_t) + \gamma^n v_\pi(s_{t+n})], \forall s_t \in \mathcal{S}, n \le H.
\end{aligned}
\tag{15}
$$

Temporal-Difference methods estimates a learning target directly from Equation 15:

$$\hat{v}_\pi(s) := \text{TD}_{\text{RL}}(n) = \underbrace{\mathbb{E}_{\pi}[\sum_{k=0}^{n-1} \gamma^k \mathcal{R}(s_t, a_t)]}_{\text{Estimated via MC Sampling}} + \underbrace{\gamma^n \hat{v}_\pi(s_{t+n})}_{\text{Bootstrapped via past estimations}}, \forall s_t \in \mathcal{S}, n \le H. \tag{16}$$

Now, we turn our attention back to our Variational Continual Learning setting. The standard VCL objective is given by Equation 2:

$$q_t(\boldsymbol{\theta}) = \arg\min_{q \in \mathcal{Q}} \mathscr{D}_{KL}(q(\boldsymbol{\theta}) \| \frac{1}{Z_t} q_{t-1}(\boldsymbol{\theta}) p(\mathcal{D}_t \mid \boldsymbol{\theta})).$$

We can similarly define a learning target as a "value" function which we aim to maximize:

$$u_{q(\boldsymbol{\theta})}(t) := -\mathscr{D}_{KL}(q(\boldsymbol{\theta}) \| \frac{1}{Z_t} q_{t-1}(\boldsymbol{\theta}) p(\mathcal{D}_t \mid \boldsymbol{\theta}))$$

$$= \mathbb{E}_{\boldsymbol{\theta} \sim q_t(\boldsymbol{\theta})} \left[ \log p(\mathcal{D}_t \mid \boldsymbol{\theta})] + \log Z_t \right] - \mathscr{D}_{KL}(q_t(\boldsymbol{\theta}) \| q_{t-1}(\boldsymbol{\theta}))$$

$$= \mathbb{E}_{\boldsymbol{\theta} \sim q_t(\boldsymbol{\theta})} \left[ \log p(\mathcal{D}_t \mid \boldsymbol{\theta})] + \log Z_t \right] - \mathscr{D}_{KL}(q_t(\boldsymbol{\theta}) \| \frac{1}{Z_{t-1}} q_{t-2}(\boldsymbol{\theta}) p(\mathcal{D}_{t-1} \mid \boldsymbol{\theta}))$$

$$= \mathbb{E}_{\boldsymbol{\theta} \sim q_t(\boldsymbol{\theta})} \left[ \log p(\mathcal{D}_t \mid \boldsymbol{\theta})] + \log Z_t \right] + u_{q(\boldsymbol{\theta})}(t-1)$$

$$= \mathbb{E}_{\boldsymbol{\theta} \sim q_t(\boldsymbol{\theta})} \left[ \sum_{i=0}^{n-2} \log p(\mathcal{D}_{t-i} \mid \boldsymbol{\theta})] + \sum_{i=0}^{n-2} \log Z_{t-i} \right] + u_{q(\boldsymbol{\theta})}(t-n+1), n \in \mathbb{N}_0, n \leq t.$$

$$(17)$$

Similarly to the RL case, it follows that $q_t(\boldsymbol{\theta}) = \arg\max_{q \in \mathcal{Q}} u_{q(\boldsymbol{\theta})}(t)$. Lastly, we assume the following estimation of the "value" function defined in Equation 17:

$$\hat{u}_{q(\boldsymbol{\theta})}(t) = \mathbb{E}_{\boldsymbol{\theta} \sim q_t(\boldsymbol{\theta})} \left[ \sum_{i=0}^{n-2} \log p(\mathcal{D}_{t-i} \mid \boldsymbol{\theta})] + \sum_{i=0}^{n-2} \log Z_{t-i} \right] + \hat{u}_{q(\boldsymbol{\theta})}(t-n+1)$$

$$= \underbrace{\mathbb{E}_{\boldsymbol{\theta} \sim q_t(\boldsymbol{\theta})} \left[ \sum_{i=0}^{n-1} \log p(\mathcal{D}_{t-i} \mid \boldsymbol{\theta})] \right]}_{\text{Estimated via MC Sampling}} - \underbrace{\mathscr{D}_{KL}(q_t(\boldsymbol{\theta}) \| q_{t-n}(\boldsymbol{\theta}))}_{\text{Bootstrapped via past posterior estimations}} + \underbrace{\left[ \sum_{i=0}^{n-1} \log Z_{t-i} \right]}_{\text{Constant w.r.t } \boldsymbol{\theta}}.$$

$$(18)$$

We notice that $Z_t$ is constant with respect to $\boldsymbol{\theta}$, hence we can disregard it and still have the same learning target. Thus, we have:

$$q_t(\boldsymbol{\theta}) = \arg\max_{q \in \mathcal{Q}} \hat{u}_{q(\boldsymbol{\theta})}(t)$$

$$= \arg\max_{q \in \mathcal{Q}} \mathbb{E}_{\boldsymbol{\theta} \sim q_t(\boldsymbol{\theta})} \left[ \sum_{i=0}^{n-1} \log p(\mathcal{D}_{t-i} \mid \boldsymbol{\theta})] \right] - \mathscr{D}_{KL}(q_t(\boldsymbol{\theta}) \| q_{t-n}(\boldsymbol{\theta})) + \left[ \sum_{i=0}^{n-1} \log Z_{t-i} \right]$$

$$= \arg\max_{q \in \mathcal{Q}} \underbrace{\mathbb{E}_{\boldsymbol{\theta} \sim q_t(\boldsymbol{\theta})} \left[ \sum_{i=0}^{n-1} \log p(\mathcal{D}_{t-i} \mid \boldsymbol{\theta})] \right] - \mathscr{D}_{KL}(q_t(\boldsymbol{\theta}) \| q_{t-n}(\boldsymbol{\theta}))}_{\text{TD}_{\text{CL}}(n)}.$$

$$(19)$$

Equation 19 is exactly n-Step Temporal-Difference target in Definition 4.3 from Section 4. The main differences from the CL recursion in Equation 17 and the RL one in Equation 15 are two-fold. First, the CL setup is not discounted (or, equivalently, assumes the discount factor $\gamma = 1$). Second, the RL recursion looks over future timesteps, while the CL one looks over past timesteps. Besides these two differences, both scenarios are strongly connected. Particularly, they share the same purpose for leveraging TD targets: to strike a balance between MC estimation (which incurs variance) and bootstrapping (which incurs bias) while estimating the learning objective.

## D  TD($\lambda$)-VCL IS A DISCOUNTED SUM OF N-STEP TD TARGETS

In Section 4, we mention that the TD-VCL learning target is a compound update that averages n-step temporal-difference targets, as per Proposition 4.4, which we prove below.

**Proposition 4.4.** $\forall n \in \mathbb{N}_0$, $n \leq t$, the objective in Equation 2 can be equivalently represented as:

$$q_t(\boldsymbol{\theta}) = \arg\max_{q \in \mathcal{Q}} \mathrm{TD}_t(n), \tag{7}$$

with $\mathrm{TD}_t(n)$ as in Definition 4.3. Furthermore, the objective in Equation 5 can also be represented as:

$$q_t(\boldsymbol{\theta}) = \arg\max_{q \in \mathcal{Q}} \frac{1-\lambda}{1-\lambda^n} \underbrace{\left[ \sum_{k=0}^{n-1} \lambda^k \mathrm{TD}_t(k+1)) \right]}_{\text{Discounted sum of TD targets}}. \tag{8}$$

*Proof.* We start by proving the equivalence between Equation 2 and Equation 7:

$$
\begin{aligned}
q_t(\boldsymbol{\theta}) &= \arg\min_{q \in \mathcal{Q}} \mathscr{D}_{KL}(q(\boldsymbol{\theta}) \,\|\, \frac{1}{Z_t} q_{t-1}(\boldsymbol{\theta}) p(\mathcal{D}_t \mid \boldsymbol{\theta})) \\
&= \arg\min_{q \in \mathcal{Q}} \mathscr{D}_{KL}(q(\boldsymbol{\theta}) \,\|\, \frac{1}{\prod_{i=0}^{n-1} Z_{t-i}} q_{t-n}(\boldsymbol{\theta}) \prod_{i=0}^{n-1} p(\mathcal{D}_{t-i} \mid \boldsymbol{\theta})) \\
&= \arg\max_{q \in \mathcal{Q}} \mathbb{E}_{\boldsymbol{\theta} \sim q_t(\boldsymbol{\theta})} \left[ \sum_{i=0}^{n-1} \log p(\mathcal{D}_{t-i} \mid \boldsymbol{\theta})] \right] - \mathscr{D}_{KL}(q_t(\boldsymbol{\theta}) \,\|\, q_{t-n}(\boldsymbol{\theta})) \\
&= \arg\max_{q \in \mathcal{Q}} \mathrm{TD}_t(n).
\end{aligned}
\tag{20}
$$

Now, we show that Equation 5 is a discounted sum of n-Step targets:

$$
\begin{aligned}
q_t(\boldsymbol{\theta}) &= \arg\max_{q \in \mathcal{Q}} \frac{1-\lambda}{1-\lambda^n} \Big[ \mathbb{E}_{\boldsymbol{\theta} \sim q_t(\boldsymbol{\theta})}[\log p(\mathcal{D}_t \mid \boldsymbol{\theta}) - \mathscr{D}_{KL}(q_t(\boldsymbol{\theta}) \,\|\, q_{t-1}(\boldsymbol{\theta}))] \\
&\qquad\qquad + \lambda \mathbb{E}_{\boldsymbol{\theta} \sim q_t(\boldsymbol{\theta})}[\log p(\mathcal{D}_t \mid \boldsymbol{\theta}) + \log p(\mathcal{D}_{t-1} \mid \boldsymbol{\theta})] - \lambda \mathscr{D}_{KL}(q_t(\boldsymbol{\theta}) \,\|\, q_{t-2}(\boldsymbol{\theta})) + \ldots \\
&\qquad\qquad + \lambda^{n-1} \mathbb{E}_{\boldsymbol{\theta} \sim q_t(\boldsymbol{\theta})}[\sum_{i=0}^{n-1} \log p(\mathcal{D}_{t-i} \mid \boldsymbol{\theta})] - \lambda^{n-1} \mathscr{D}_{KL}(q_t(\boldsymbol{\theta}) \,\|\, q_{t-n}(\boldsymbol{\theta})) \Big] \\
&= \arg\max_{q \in \mathcal{Q}} \frac{1-\lambda}{1-\lambda^n} \Big[ \mathrm{TD}_t(1) + \lambda \mathrm{TD}_t(2) + \ldots \lambda^{n-1} \mathrm{TD}_t(n) \Big] \\
&= \arg\max_{q \in \mathcal{Q}} \frac{1-\lambda}{1-\lambda^n} \underbrace{\left[ \sum_{k=0}^{n-1} \lambda^k \mathrm{TD}_t(k+1)) \right]}_{\text{Disconted sum of TD targets}}.
\end{aligned}
\tag{21}
$$

$\square$

In Equation 7, if we set $n = 1$, the n-Step TD target recovers the VCL objective. Furthermore, it is worth highlighting that an n-Step TD target is **not** the same as n-Step KL Regularization. The latter leverages several previous posterior estimates, while the former only relies on a single estimate. Lastly, we can follow a similar idea to prove that the n-Step KL Regularization objective is a simple average of n-step TD targets, by leveraging the expansion in Equation 9 and identifying the sum of TD targets.

# E  TD-VCL: A SPECTRUM OF CONTINUAL LEARNING ALGORITHMS

In this Section, we describe how TD-VCL spans a spectrum of algorithms that mix different levels of Monte Carlo approximation for expected log-likelihood and KL regularization. Our goal is to show that by choosing specific hyperparameters for Equation 5, one may recover vanilla VCL in one extreme and n-Step KL regularization in the opposite.

Let us consider the TD-VCL objective in Equation 5:

$$\arg\max_{q\in\mathcal{Q}} \mathbb{E}_{\boldsymbol{\theta}\sim q_t(\boldsymbol{\theta})} \Big[ \sum_{i=0}^{n-1} \frac{\lambda^i(1-\lambda^{n-i})}{1-\lambda^n} \log p(\mathcal{D}_{t-i} \mid \boldsymbol{\theta}) \Big] - \sum_{i=0}^{n-1} \frac{\lambda^i(1-\lambda)}{1-\lambda^n} \mathscr{D}_{KL}(q_t(\boldsymbol{\theta}) \mid\mid q_{t-i-1}(\boldsymbol{\theta})).$$

Trivially, if we set $\lambda = 0$, assuming $0^0 = 1$, it recovers the Vanilla VCL objective, as stated in Equation 3, regardless of the choice of $n$.

More interestingly, we investigate the learning target as $\lambda \to 1$:

$$\lim_{\lambda\to 1} \Big\{ \mathbb{E}_{\boldsymbol{\theta}\sim q_t(\boldsymbol{\theta})} \Big[ \sum_{i=0}^{n-1} \frac{\lambda^i(1-\lambda^{n-i})}{1-\lambda^n} \log p(\mathcal{D}_{t-i} \mid \boldsymbol{\theta}) \Big] - \sum_{i=0}^{n-1} \frac{\lambda^i(1-\lambda)}{1-\lambda^n} \mathscr{D}_{KL}(q_t(\boldsymbol{\theta}) \mid\mid q_{t-i-1}(\boldsymbol{\theta})) \Big\}$$

$$= \mathbb{E}_{\boldsymbol{\theta}\sim q_t(\boldsymbol{\theta})} \Big[ \sum_{i=0}^{n-1} \underbrace{\lim_{\lambda\to 1} \Big\{ \frac{\lambda^i(1-\lambda^{n-i})}{1-\lambda^n} \Big\}}_{(I)} \log p(\mathcal{D}_{t-i} \mid \boldsymbol{\theta}) \Big] - \sum_{i=0}^{n-1} \underbrace{\lim_{\lambda\to 1} \Big\{ \frac{\lambda^i(1-\lambda)}{1-\lambda^n} \Big\}}_{(II)} \mathscr{D}_{KL}(q_t(\boldsymbol{\theta}) \mid\mid q_{t-i-1}(\boldsymbol{\theta}))$$

Let us develop (I) and (II) separately by applying the L'Hôpital's rule. First, for (I):

$$\lim_{\lambda\to 1} \Big\{ \frac{\lambda^i(1-\lambda^{n-i})}{1-\lambda^n} \Big\} = \lim_{\lambda\to 1} \Big\{ \frac{i\lambda^{i-1}(1-\lambda^{n-i}) - \lambda^i(n-i)\lambda^{n-i-1}}{-n\lambda^{n-1}} \Big\}$$
$$= \lim_{\lambda\to 1} \Big\{ \frac{i\lambda^{i-1} - i\lambda^{n-1} - (n-i)\lambda^{n-1}}{-n\lambda^{n-1}} \Big\} = \frac{n-i}{n}. \tag{22}$$

Now, for (II):

$$\lim_{\lambda\to 1} \Big\{ \frac{\lambda^i(1-\lambda)}{1-\lambda^n} \Big\} = \lim_{\lambda\to 1} \Big\{ \frac{i\lambda^{i-1}(1-\lambda) - \lambda^i}{-n\lambda^{n-1}} \Big\} = \frac{1}{n}. \tag{23}$$

Applying Equations 22 and 23 to TD-VCL objective, we obtain:

$$\arg\max_{q\in\mathcal{Q}} \mathbb{E}_{\boldsymbol{\theta}\sim q_t(\boldsymbol{\theta})} \Big[ \sum_{i=0}^{n-1} \frac{(n-i)}{n} \log p(\mathcal{D}_{t-i} \mid \boldsymbol{\theta}) \Big] - \sum_{i=0}^{n-1} \frac{1}{n} \mathscr{D}_{KL}(q_t(\boldsymbol{\theta}) \mid\mid q_{t-i-1}(\boldsymbol{\theta})),$$

which is exactly the N-Step KL Regularization objective.

# F   IMPLEMENTATION DETAILS

**Operationalization.** For all experiments, we use a Gaussian mean-field approximate posterior and assume a Gaussian prior $\mathcal{N}(0, \sigma^2 \boldsymbol{I})$. We parameterize all distributions as deep networks. For all variational objectives, we compute the KL term analytically and employ the Monte Carlo approximations for the expected log-likelihood terms, leveraging the reparametrization trick (Kingma & Welling, 2014) for computing gradients. Lastly, we employed likelihood-tempering (Loo et al., 2021) to prevent variational over-pruning (Trippe & Turner, 2018).

**Model Architecture and Hyperpatameters**. We adopted fully connected neural networks. We chose different depths and sizes depending on the benchmark, and we provide a full list of hyperparameters in Appendix G. For training, we used the Adam optimizer (Kingma & Ba, 2015). We implemented early stopping with a patience parameter of five epochs, which drastically reduced the number of epochs needed for each new task.

We initialize the prior with variance $1e-5$. In contrast to what was reported by Nguyen et al. (2018), we found no gains in initializing the variational parameters with the maximum likelihood estimate parameters. Therefore, we started by sampling from the prior.

**Hyperparamter Tuning Protocol.** For the proposed methods, we mainly tuned three hyperparameters: $n$ (as in n-Step KL), $\lambda$ (as in TD-VCL), and $\beta$ (the likelihood tempering parameter). We conducted a grid search for each evaluated benchmark, with $n \in \{1, 2, 3, 5, 8, 10\}$, $\lambda \in \{0.0, 0.1, 0.5, 0.8, 0.9, 0.99\}$, and $\beta \in \{1e-5, 1e-4, 1e-3, 5e-3, 1e-2, 5e-2, 1e-1, 1.0\}$. For VCL, we tuned the $\beta$ hyperparameter in the same way.

**Reproducibility**. Reported results are averaged across ten different seeds. Error bars represent 95% confidence intervals, while Table 1 shows 2-sigma errors up to two decimal places. We execute all experiments using a single GPU RTX 4090. We provide our implementation code for the proposed methods (TD-VCL and n-Step), as well as considered baselines (Batch MLE, Online MLE, VCL, and VCL CoreSet) in `https://anonymous.4open.science/r/vcl-nstepkl-5707`.

# G   HYPERPARAMETERS

Table 2 provides the shared hyperparameters used in each benchmark. Tables 3 and 2 provided the specific hyperparameters for n-Step KL and TD-VCL methods, respectively.

|  | PermutedMNIST | SplitMNIST | SplitNotMNIST |
|---|---|---|---|
| **Batch Size** | 256 | 256 | 256 |
| **Max. Number of Epochs** | 100 | 100 | 100 |
| **Network Layers** | [100, 100] | [256, 256] | [150, 150, 150, 150] |
| **Number of Heads** | 1 | 5/1 | 1 |
| **Learning Rate** | 1e-3 | 1e-3 | 1e-3 |
| **Prior Variance** | 1e-5 | 1e-5 | 1e-5 |

Table 2: Training hyperparameters. These are shared across all evaluated methods.

|  | PermutedMNIST | SplitMNIST | SplitNotMNIST |
|---|---|---|---|
| $n$ | 5 | 4 | 5 |
| $\beta$ | 5e-3 | 5e-2 | 5e-2 |

Table 3: Hyperparameters for the n-Step KL Regularization method.

|  | PermutedMNIST | SplitMNIST | SplitNotMNIST |
|---|---|---|---|
| $n$ | 8 | 4 | 3 |
| $\lambda$ | 0.5 | 0.8 | 0.1 |
| $\beta$ | 1e-3 | 5e-2 | 1e-3 |

Table 4: Hyperparameters for the TD-VCL method.

## H    SplitMNIST: Additional Results

### H.1    Multi-Head Network

Figure 7 presents the results for the SplitMNIST benchmark. The first five plots bring the results per each task, while the last shows the average accuracy as the number of observed tasks grows. As the results suggest, SplitMNIST with multi-head networks is the easiest benchmark: even Online MLE presents a final average accuracy of around 90%. All variational methods present almost perfect results, ranging between 97% and 98%. As stated in Section 5.1, these positive results are a consequence of using multi-head networks, which end up training different classifiers on top of the same backbone. This architecture would naturally disregard negative transfer and, therefore, Catastrophic Forgetting. They also assume *a priori* knowledge about the number of tasks, which is unrealistic for CL settings.

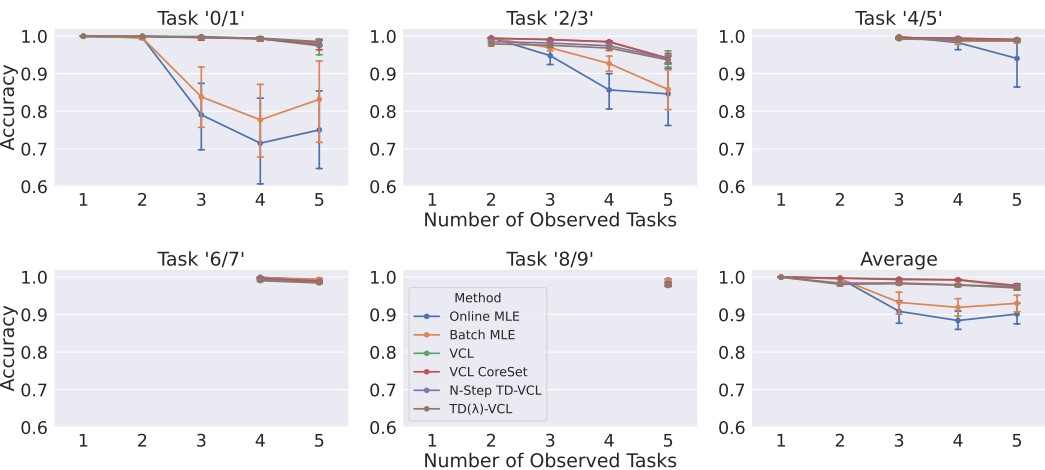

Figure 7: **SplitMNIST results with multi-head networks**. The first five plots show results per task, and the last one is an average across tasks. As a consequence of multi-head networks trivializing the Continual Learning challenge, all methods attain high accuracy. In particular, variational methods accuracies ranging from 97% and 98%.

### H.2    Single-Head Network

In this Section, We re-evaluate the SplitMNIST benchmark by only allowing single-head architectures.Figure 8 presents the results. As expected, the performance of all methods drops substantially, as the problem of representing a streaming of tasks in a single classifier is much harder. However, we highlight that n-Step KL and TD-VCL presented slightly better results than VCL and VCL CoreSet, demonstrating again the effectiveness of the proposed learning objectives.

Interestingly, the average accuracy does not decrease monotonically, as usually expected due to Catastrophic Forgetting. It actually decreases substantially after Task 3 and increases back. This evidence suggests a negative transfer between tasks, particularly in Task 1 while learning Task 3, as the first plot presents. We believe this could be addressed with more expressive architectures that better disentangle features and avoid negative transfer. Nonetheless, this is outside our scope, as we focus on studying the effect of Catastrophic Forgetting in Continual Learning.

## I    SplitNotMNIST: Additional Results

In this Section, we evaluate methods in SplitNotMNIST by considering single-head networks for all methods and show per-task performance. As highlighted in Section 5.2, SplitNotMNIST is a considerably harder benchmark, and the choice of simpler deep architectures naturally results in

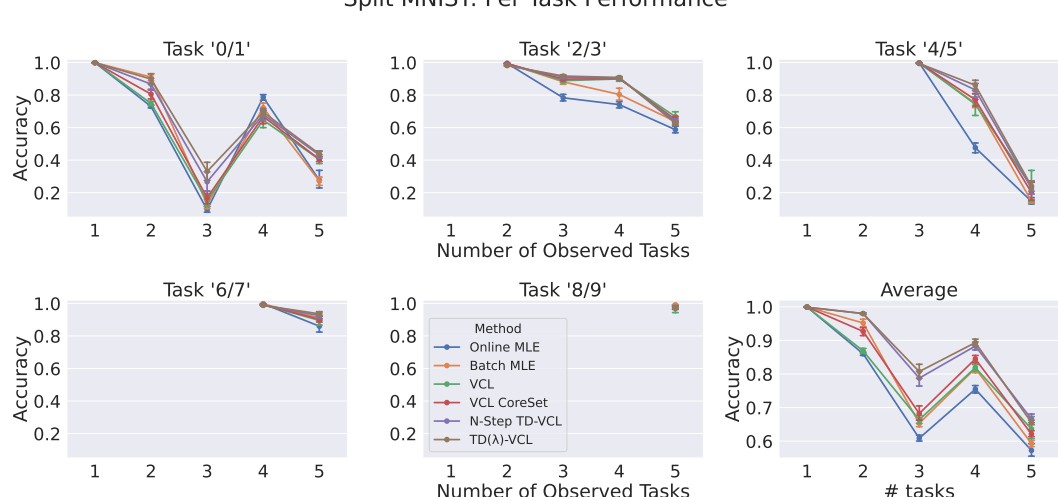

Figure 8: **SplitMNIST results with single-head networks**. In this more robust evaluation setting, tasks are enforced to share all learned parameters. Consequently, the effect of Catastrophic Forgetting (and task negative transfer) is explicit. TD-VCL objectives present slightly better average accuracy across tasks in comparison with standard VCL variants.

higher approximation errors. Our goal is to evaluate how the presented methods behave under this circumstance.

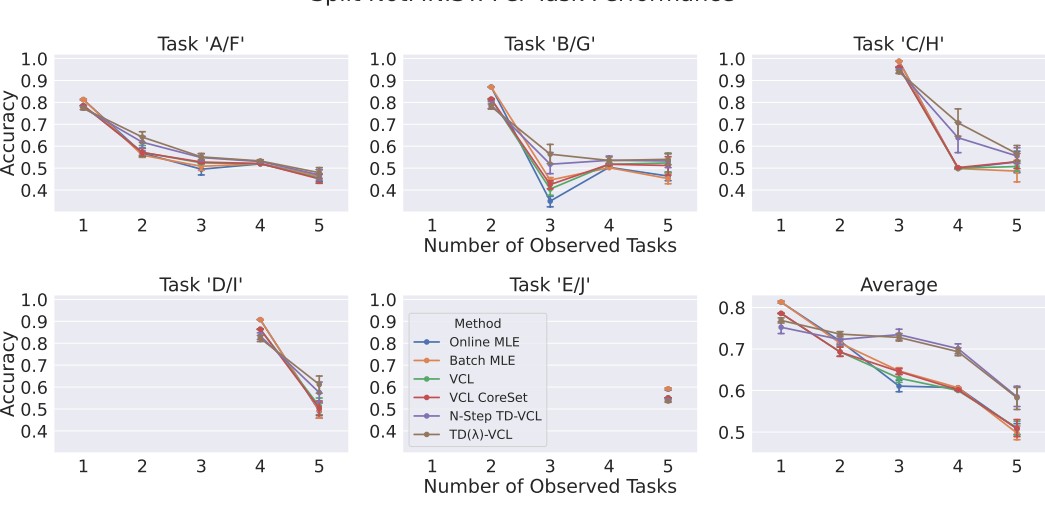

Figure 9: **SplitNotMNIST results**. The first five plots show results per task, and the last one is an average across them. SplitNotMNIST is considerably harder to fit with modest deep architectures, leading to a setup where posteriors induce high approximation errors. As a result, the standard VCL variants performs similarly to non-variational approaches. TD-VCL surpasses all methods and shows more robustness to Catastrophic Forgetting under this high approximation error setting.

Figure 9 presents the results. As expected, even learning the current task is challenging. This characteristic contrasts with PermutedMNIST and SplitMNIST, where all models could at least fit the current task almost perfectly. MLE methods fit the current task slightly better since their objectives are not regularized by the prior or previous posterior. However, this same reason caused them to suffer from Catastrophic Forgetting more drastically, as they tend to focus on fitting the current task and

disregard past ones. Overall, TD-VCL objectives maintained the best trade-off between plasticity and memory stability, aligning with the results in the other benchmarks.

## J  HYPERPARAMETERS ROBUSTNESS ANALYSIS

In this Section, we present robustness studies in the PermutedMNIST benchmark with respect to the relevant hyperparameters. Our goal is to evaluate how they affect the performance of the proposed methods.

### J.1  n-STEP KL REGULARIZATION

Figure 10 presents the ablation study of the n-step KL Regularization method in the PermutedM-NIST benchmark. We designed this study to highlight the two most sensitive hyperparameters: $n$, the n-step size, and $\beta$, the likelihood-tempering parameter.

Similarly to VCL, this method is sensitive to the choice of $\beta$. Higher values will prevent the model from fitting new tasks, a manifestation of variational over-pruning. On the other hand, lower values will not retain knowledge properly, suffering from Catastrophic Forgetting. Mild values (0.001, 0.005, 0.01) balanced well this trade-off.

In terms of $n$, we observe benefits of up to 5 steps. Beyond that, the effect saturates, even becoming slightly detrimental. This observation suggests that too-old posterior estimates may not be useful to retain knowledge, as compared to most recent ones.

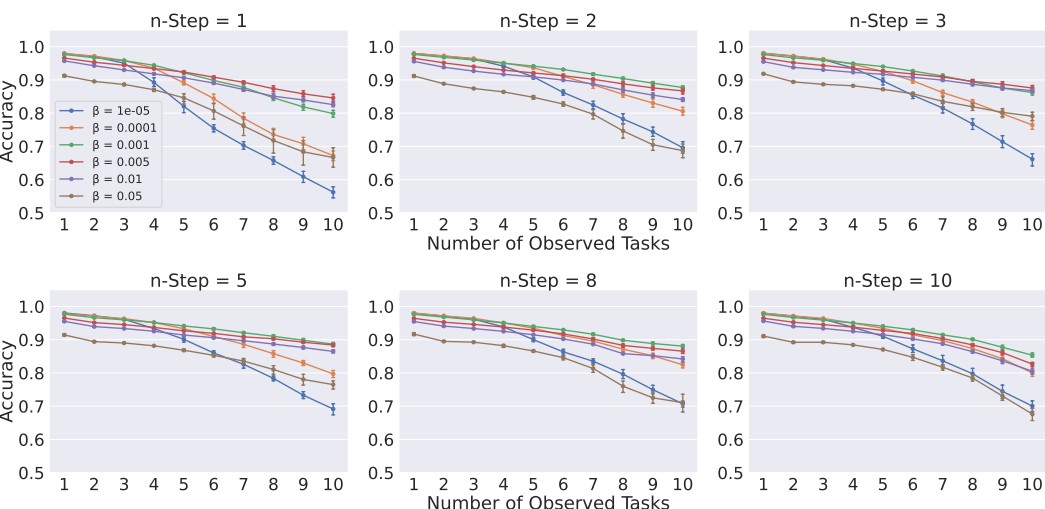

Figure 10: **Hyperparameter Robustness Analysis for n-Step KL Regularization in PermutedM-NIST.** The plots show the effect of the likelihood-tempering parameter $\beta$ for different $n$. For $\beta$, too high values negatively affect fitting new tasks, and too low values disregard the regularization of previous posteriors, leading to Catastrophic Forgetting. For $n$, we observe benefits while increasing up to $n = 5$, and the effect saturates.

### J.2  TD($\lambda$)-VCL

Figure 11 shows the ablation study for TD-VCL. For this setup, we considered a fixed value of $\beta$, as our hyperparameter search suggested the same trends for n-Step KL Regularization and TD-VCL. Hence, we simplify the analysis to consider only $n$ and $\lambda$.

TD-VCL presents mild sensitivity to the choice of $\lambda$. The effect is more pronounced as the method observes more tasks, with a slight preference for lower values for some choices of $n$. We believe that the choice of $\lambda$ will fundamentally depend on how most recent estimates are better and more

informative than old ones. In the case where they present similar approximation errors, the choice of $\lambda$ causes less impact, and, therefore, there is less difference between leveraging N-Step TD-VCL and TD($\lambda$)-VCL objectives.

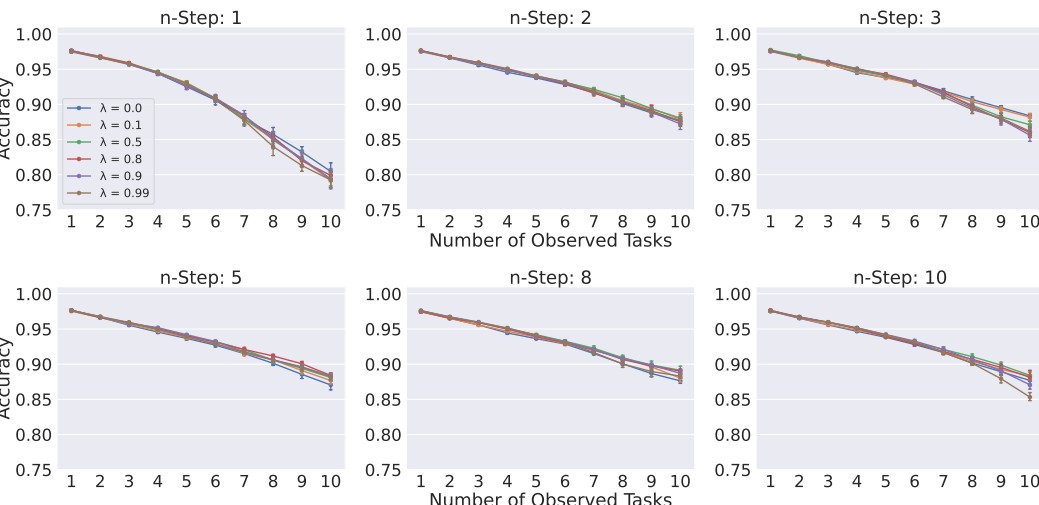

Figure 11: **Hyperparameter Robustness Analysis for TD($\lambda$)-VCL in PermutedMNIST**. The plots show the effect of $\lambda$ for different choices of $n$. The learning objective presents mild sensitivity to the choice of $\lambda$ in this benchmark, and the effect is more pronounced as the number of observed tasks increases.

