# OpenReview forum: "Temporal-Difference Variational Continual Learning"
_ICLR.cc/2025/Conference — Submitted to ICLR 2025_

### Official Review · Reviewer_awe2 · 2024-10-23

**Soundness:** 3
**Presentation:** 3
**Contribution:** 3
**Rating:** 5
**Confidence:** 4

**Summary:**

The paper focuses on mitigating the issue of cumulative error accumulation in variational continual learning due to relying on a single posterior from the past task. The paper formulates n-Step KL-VCL, which allows for regularizing network updates using past n posteriors. In doing so, it formulates the likelihood term to integrate replay samples from past n tasks. Furthermore, it proposes TD($\lambda$)-VCL, which connects variational continual learning with TD methods from reinforcement learning.

**Strengths:**

1. The paper makes a significant contribution by drawing on Temporal-Difference methods to mitigate error accumulation in variational continual learning. Thus proposed formulation allows the regularization using past $n$ posteriors and incorporation of a replay buffer for previous $n$ tasks into the principled framework of variational continual learning.
2. The experiments show a performance boost compared to the baselines. The propositions and their proofs further enhance the strength of this work.
3. The paper is well-organized and easy to follow. The authors provide a thorough analysis of their method on benchmark datasets, along with sensitivity analysis of hyper-parameters.

**Weaknesses:**

1. One major weakness is that the benchmarks include small-scale MNIST variants (permuted MNIST and single-headed MNIST/not-MNIST tasks) only.
2. The benchmarks are constrained to the task-incremental learning, where the task identifier is provided during prediction. The paper's claim of effort to raise the standards for evaluating continual learning is not strong, as recent works commonly focus on the more challenging class-incremental learning setting, which doesn't require task identifiers for prediction.

I would be happy to raise the score if these weaknesses and the following questions are addressed.

**Questions:**

1. As most recent works on Bayesian continual learning [1,2] experiment with CIFAR and tiny ImageNet, it would be interesting to see the results when applied to such relatively more complex datasets.
2. Since the proposed method incorporates a replay buffer, it would be interesting to see how it compares in a class-incremental learning setting against replay-based methods like ER [3].

[1] Kumar, A., Chatterjee, S., Rai, P. (2021). Bayesian Structural Adaptation for Continual Learning. In Proceedings of the 38th International Conference on Machine Learning (pp. 5850–5860). PMLR.

[2] Thapa, J., Li, R. (2024). Bayesian Adaptation of Network Depth and Width for Continual Learning. In Forty-first International Conference on Machine Learning.

[3] Arslan Chaudhry, Marcus Rohrbach, Mohamed Elhoseiny, Thalaiyasingam Ajanthan, Puneet K Dokania, Philip HS Torr, and Marc’Aurelio Ranzato. Continual learning with tiny episodic memories. arXiv preprint arXiv:1902.10486, 2019.

---

### Official Review · Reviewer_vxsa · 2024-10-30

**Soundness:** 3
**Presentation:** 3
**Contribution:** 2
**Rating:** 6
**Confidence:** 5

**Summary:**

The authors introduce the Variational Continual Learning (VCL) paper, which is a Bayesian CL  approach where posterior distributions are updated recursively, highlighting the compounding effect of VCL and error accumulation due to objective depending on the posterior of an immediate previous task. To address this, the paper proposes two main solutions. First, they introduce an n-step KL regularization objective, which incorporates multiple past posterior estimates. This approach reduces the impact of individual errors and enhances the overall reliability of the model. Additionally, the authors draw parallels between their approach and temporal-difference (TD) methods from reinforcement learning – no experiment in RL though. They suggest that integrating concepts from TD learning can further improve learning outcomes by providing a more robust way to handle updates. The proposed methods were validated through experiments against standard VCL techniques and non-variational baselines, using well-known CL benchmarks. The paper also presents detailed theoretical insights to validate the claims made. The results showed improved performance, effectively mitigating the problem of catastrophic forgetting. This research offers valuable insights into developing more robust continual learning frameworks by combining variational inference with temporal-difference learning mechanisms. It would be more interesting to see the results with the larger model on complex datasets

**Strengths:**

1. **Addressed a Potential Gap in Current Bayesian Continual Learning**:
   The proposed method effectively addresses the issue of Catastrophic Forgetting by utilizing multiple past posterior estimates, which helps to dilute the impact of individual errors that could compound over time.

2. **Enhanced Learning Objectives**:
   By integrating n-Step KL regularization, the model can leverage a broader context from previous tasks, leading to improved performance in continual learning scenarios compared to standard Variational Continual Learning (VCL) methods.

3. **Single-Step Optimization**:
   Unlike some existing methods that require complex two-step optimizations or replay mechanisms, this approach simplifies the learning process by naturally incorporating replay into the learning objective.

**Weaknesses:**

## Key Points of Consideration

### 1. Dependence on Hyperparameter Tuning
- **Effectiveness Contingency**: The performance of n-Step KL regularization is heavily dependent on the appropriate setting of its hyperparameters.

### 2. Increased Computational Complexity
- **Robustness vs. Overhead**: While utilizing multiple past estimates can enhance robustness, it may introduce significant computational overhead, particularly in resource-limited environments.
- **Training and Inference Time**: It is essential to report training and inference times, as Bayesian models are generally slower compared to deterministic counterparts.

### 3. Assumption of IID Tasks
- **Real-World Applicability**: The framework operates under the assumption that tasks are independent and identically distributed (IID). This assumption may not hold in many real-world scenarios, potentially limiting the framework's applicability.

### 4. Potential for Bias in Estimates
- **Impact of Biased Estimates**: If earlier posterior estimates are significantly biased, they could adversely affect the learning target, even with proposed mitigation strategies.

### 5. Scalability of the Bayesian Framework
- **Applicability Limitations**: Focusing on a Bayesian approach may restrict applicability to other models or frameworks that do not align with Bayesian principles. The framework may struggle with complex datasets exhibiting multiple distribution shifts, such as CIFAR10/100 and ImageNet, especially when utilizing larger architectures like ResNets and ViTs.

### 6. Limited Experiments
- **Validation Scope**: The framework has only been validated on MNIST and its variations and compared solely with the VCL paper. There are other prominent Bayesian continual learning works based on Mean-Field Variational Inference (MVFI), such as UCB [1], UCL [2], and Bayesian Structural Adaptation [3]. It would be beneficial to evaluate these frameworks after applying dilation techniques.
- **Lack of Analysis**: The main section claims contributions, but there is a lack of empirical analysis in the results section for RL.

## Contribution to Literature
Despite its limitations, the work presents a valuable contribution to the existing literature on continual learning.

## Questions for Further Clarification
1. **Learning Strategy**: For SplitMNIST and SplitNotMNIST, which learning strategy was employed? Was it Task-Incremental Learning (TIL) or Class-Incremental Learning (CIL)?
2. **Re-weighting Posteriors**: What is the intuition behind re-weighting the posteriors with KL-divergence to mitigate error accumulation? What are the implications when \( n = t \)?
3. **Exemplar-Free Setting**: How does the framework perform in an exemplar-free setting?

I will be happy to increase the score if the authors show empirical validation that the framework is scalable to larger models and complex datasets
### References
[1] Ahn, Hongjoon, et al. "Uncertainty-based continual learning with adaptive regularization." Advances in neural information processing systems 32 (2019).

[2] Ebrahimi, Sayna, et al. "Uncertainty-guided continual learning in Bayesian neural networks–Extended abstract." Proc. IEEE Conf. Comput. Vis. Pattern Recognition (CVPR). 2018.

[3] Kumar, Abhishek, Sunabha Chatterjee, and Piyush Rai. "Bayesian structural adaptation for continual learning." International Conference on Machine Learning. PMLR, 2021.

**Questions:**

### Please refer to the weakness

---

### Official Review · Reviewer_DJ93 · 2024-11-04

**Soundness:** 2
**Presentation:** 3
**Contribution:** 1
**Rating:** 3
**Confidence:** 3

**Summary:**

This paper introduces TD-VCL, aiming to mitigate Catastrophic Forgetting in continual learning (CL) by using a variational framework inspired by reinforcement learning’s temporal-difference (TD) methods.

**Strengths:**

The paper is well-written and easily understandable.

**Weaknesses:**

This work builds on an earlier approach to variational continual learning. While applying a temporal modification to the variational objective to mitigate model drift is intuitive, and drawing a connection to reinforcement learning is conceptually interesting, this work and its benchmarks feel largely disconnected from recent advances in continual learning. Had this work been published six years ago, it might have been more impactful, but recent developments have rendered variational models less relevant due to their limitations in scalability and stability.

The experiments are confined to benchmarks like PermutedMNIST, SplitMNIST, and SplitNotMNIST—datasets that are relatively simple and fall short of reflecting real-world continual learning challenges. More recent works typically include larger and more complex datasets such as CIFAR-100 and ImageNet, which would provide a more realistic evaluation of the method.

Additionally, the paper’s evaluation lacks comparisons to newer, stronger baselines in the field. While standard VCL and its variants are included, recent advanced methods, such as ALTA, DER, and L2P, are absent. This omission raises questions about the practical relevance and competitiveness of the proposed method.

**Questions:**

To improve the impact of the method, the authors could consider building on more recent models and benchmarks or even integrating connections to neural science, potentially aligning the method more closely with the evolving landscape of continual learning.

---

### Official Review · Reviewer_Z6hC · 2024-11-04

**Soundness:** 2
**Presentation:** 2
**Contribution:** 2
**Rating:** 3
**Confidence:** 5

**Summary:**

In this paper, the authors propsoed a new version of variational continual learning (VCL) which combines n-step regularization loss with temporal difference. The n-step loss considers all posterior and log likelihood before n steps, and the distribution that minimizes the n-step loss can cover all n tasks. As an improved version, TD($\lambda$)-VCL uses the weighted sum of the log likelihood and KL regularization, and controls the weights using $\lambda$. In the experiment, TD($\lambda$)-VCL achieves better performance than other baselines in variation of MNIST expeirments.

**Strengths:**

Strengths

1. In VCL or VCL variants, they formulate the KL regularization loss only using the posterior distribution on previous task. However, in this paper, the scheme that using all the n posteriors before n-steps has strong advantage for tackling the catastrophic forgetting.

**Weaknesses:**

Weaknesses

1. To minimize Eq.(8), we should store both the memory buffer and the posterior distribution on previous tasks. However, I think that this scheme takes large memory, and higly inefficient. Most of the VCL variants (UCL[1] or UCB[2]) only stores the posterior distribution of previous task and also outperform the VCL and other baselines.

2. The authors should include other baselines ([1]. [2], and other regularization based CL methods). In the PermutedMNIST or Split MNIST experiment, the overall accuracy is too low. In [1] and [2], they achieves much better performance than the proposed methods without using large amount of memory. Therefore, I think the contribution on TD($\lambda$)-VCL is too weak

3. To strengthen the effectiveness of TD($\lambda$)-VCL, I think the experiments on using CNN architecture with larger dataset should be carried out. I think the algorithms that are applied only at a small scale scenario does not have any advantage these days.



[1] Ahn et.al., Uncertainty-based Continual Learning with Adaptive Regularization, NeurIPS 2019

[2] Ebrahimi et. al., Uncertainty-guided Continual Learning with Bayesian Neural Networks, ICLR, 2020

**Questions:**

Already mentioned in weaknesses section

---

### Author Response · Authors · 2024-11-28
**Thank you for your reviews**

Dear reviewers,

We would like to extend our sincere gratitude for your time and feedback to improve our paper. We understand that the reviewers agreed that our work requires more empirical validation, particularly in more complex benchmarks and comparing against other baselines. Despite our efforts, we did not have enough time to implement and run all experiments we believed that would address the concerns raised. We also noticed some misunderstandings of the adopted evaluation setup, which would also require further clarifications in the paper content.

Given these circumstances, we decided not to engage in the rebuttal discussion without these required changes. Nonetheless, we would like to leave this message here to thank the reviewers for their time and useful feedback.

---

### Meta-Review · Area_Chair_cchu · 2024-12-20

**Metareview:**

This paper draws links between new Variational Continual Learning methods and Temporal-Difference mthods.

This meta-review is relatively short as all reviewers agreed that the paper requires more experiments to verify the claims made in the paper (MNIST scale experiments are not enough now, even if it was when the original VCL paper came out years ago). The authors wrote a short response acknowledging this limitation in the current version (and some ways to improve clarity / reduce misunderstandings), and look forward to a future version incorporating the reviewers' feedback.

Minor point: Reviewer Z6hC also brings up that the accuracies seem low. Looking into this, I agree that the accuracies are surpisingly low, eg for Permuted MNIST and VCL (Table 1). Nguyen et al. (2018) have higher accuracies, so this might be worth looking into / actively addressing reasons for in a future version of the paper. Swaroop et al. (2019) improve VCL performance further, as does Generalised VCL (Loo et al., 2020).

**Additional Comments On Reviewer Discussion:**

The authors understandably chose not to engage in a detailed rebuttal, and I look forward to a future version of the paper.

---

### Decision · Program_Chairs · 2025-01-22

Reject